# Improving the Statistical Efficiency of Cross-Conformal Prediction

**Matteo Gasparin** [1]   **Aaditya Ramdas** [2]

## Abstract

Vovk (2015) introduced cross-conformal prediction, a modification of split conformal designed to improve the width of prediction sets. The method, when trained with a miscoverage rate equal to $\alpha$ and $n \gg K$, ensures a marginal coverage of at least $1 - 2\alpha - 2(1-\alpha)(K-1)/(n+K)$, where $n$ is the number of observations and $K$ denotes the number of folds. A simple modification of the method achieves coverage of at least $1 - 2\alpha$. In this work, we propose new variants of both methods that yield smaller prediction sets without compromising the latter theoretical guarantees. The proposed methods are based on recent results deriving more statistically efficient combination of p-values that leverage exchangeability and randomization. Simulations confirm the theoretical findings and bring out some important tradeoffs.

## 1. Introduction

Conformal prediction has emerged as a general and versatile framework for constructing prediction sets in regression and classification tasks (Shafer & Vovk, 2008). Unlike traditional methods, which often depend on rigid distributional assumptions, conformal prediction transforms point predictions from any prediction (or black-box) algorithm into prediction sets that guarantee valid finite-sample marginal coverage. Originally introduced by Saunders et al. (1999), it has become increasingly influential, with numerous methods and extensions being proposed since its introduction.

In particular, full conformal prediction by Vovk et al. (2005), demonstrates favorable properties regarding the coverage and the size of the prediction set. However, these advantages are counterbalanced by a substantial computational cost, which limits its practical application. In fact, the method requires one to train the model for every possible value of the response, and this procedure is usually computationally burdensome. To alleviate this problem, split conformal prediction (Papadopoulos et al., 2002; Lei et al., 2018) has been proposed as a solution. The procedure involves a random partition of the data into two subsets: the first subset is used to train the prediction algorithm, while the remaining part is used to calibrate the predictions and to obtain the prediction interval. Although this variant proves to be computationally efficient, it suffers from reduced efficiency in terms of the width of the resulting prediction set; this is due to the fact that only a fraction of the data is used to train the model.

Several "hybrid" solutions have been proposed in the literature, which can be considered between split conformal prediction and full conformal prediction. Examples include cross-conformal prediction (Vovk, 2015; Vovk et al., 2018), multi-split conformal prediction (Solari & Djordjilović, 2022), the jackknife+ (Barber et al., 2021) and out-of-bag conformal prediction (Linusson et al., 2020; Gupta et al., 2022). These techniques generally result in smaller prediction intervals compared to split conformal prediction and involve less computational effort than full conformal prediction. However, one of the main drawbacks of these methods is the reduced marginal coverage guarantee, which is less than the usual $1 - \alpha$ level.

In this work, we focus the attention on cross-conformal prediction and we prove that the method can be improved without altering the coverage guarantee. In other words, we are able to obtain smaller prediction sets while ensuring the same (worst-case) miscoverage rate. Starting from a modification of the method (Vovk et al., 2018; Barber et al., 2021), the new results are obtained using recent findings on the combination of dependent p-values derived in Gasparin et al. (2025). Importantly, these results are obtained in a fully general manner, and do not need any specific prediction model or ensemble method to be used.

The structure of the paper is as follows. In Section 2 we illustrate the problem setup and related work. In Section 3 cross-conformal prediction is described while the new methods and results are presented in Section 4. Section 5 presents some empirical results. In particular, Section 5.1 contains some simulation results, while an application to a real-world dataset is presented in Section 5.2.

[1]Department of Statistical Sciences, University of Padova, Padua, Italy [2]Department of Statistics and Data Science, Carnegie Mellon University, Pittsburgh, PA, USA. Correspondence to: Matteo Gasparin <matteo.gasparin.1@phd.unipd.it>.

*Proceedings of the $42^{nd}$ International Conference on Machine Learning*, Vancouver, Canada. PMLR 267, 2025. Copyright 2025 by the author(s).

## 2. Problem Setup and Related Work

Assume we have independent and identically distributed (iid) training samples $Z_i = (X_i, Y_i) \in \mathcal{X} \times \mathcal{Y}$, $i = 1, \ldots, n$, drawn from a probability distribution $Q$, where $\mathcal{X}$ represents the feature space and $\mathcal{Y}$ the response space. Using these training data, our goal is to obtain a prediction set for the response variable $Y_{n+1}$ based on the covariates $X_{n+1}$, under the assumption that the test pair $(X_{n+1}, Y_{n+1})$ is independently sampled from the same distribution $Q$. In what follows, the results will be shown to hold more generally under the assumptions of exchangeability of the $n+1$ data points with the iid assumption as a special case. A typical scenario involves applying a prediction algorithm to the training data in order to find a prediction for the response value. In particular, let $\hat{\mu} : \mathcal{X} \to \mathcal{Y}'$ be a regression function obtained by applying an algorithm $\mathcal{A}$ to the training points, where $\mathcal{Y}'$ is the prediction space (in regression problems we usually have $\mathcal{X} = \mathbb{R}^p$ and $\mathcal{Y} = \mathcal{Y}' = \mathbb{R}$). Formally, $\mathcal{A}$ is a mapping from $\cup_{d \geq 1} (\mathcal{X} \times \mathcal{Y})^d$ (the set of all possible training datasets of any size $d \geq 1$), to the space of functions $\mathcal{X} \to \mathcal{Y}'$. Starting from the regression function $\hat{\mu}$, we aim to construct a prediction set $\hat{C}(X_{n+1})$ that contains the point $Y_{n+1}$ with high probability. Since no assumptions are made about the distribution $Q$, the method is said to be *distribution-free*.

Before proceeding with the remainder of the paper, we define the score function $s = s((x, y); \mathcal{D})$, which quantifies the non-conformity of a point in the sample space with respect to the dataset $\mathcal{D} \in (\mathcal{X} \times \mathcal{Y})^d$ used to train the prediction model. In particular, we assume that the score function $s$ adheres to a symmetry property:

$$s\big((x, y); \mathcal{D}\big) = s\big((x, y); \mathcal{D}^\pi\big), \tag{1}$$

where $\pi$ is any permutation of the indices $[d] := \{1, \ldots, d\}$ and $\mathcal{D}^\pi$ refer to the dataset whose elements are permuted by $\pi$. For example, when considering residual scores $|y - \hat{\mu}(x)|$ in a regression problem, the symmetry of the score function is satisfied if the prediction algorithm is symmetric, which means that $\mathcal{A}(\mathcal{D}) = \mathcal{A}(\mathcal{D}^\pi)$. In addition, we denote the dataset containing the observations in the set $I$ as $\mathcal{D}_I = (Z_i : i \in I)$.

### 2.1. Related Work

As outlined in the Introduction, conformal prediction was first introduced and formalized by Saunders et al. (1999) and Vovk et al. (2005). Several influential contributions to the framework include the works of Lei et al. (2018), Romano et al. (2019), and Barber et al. (2021). Extensions and generalizations of the methods have been proposed by Kim et al. (2020) and Gupta et al. (2022), among others. Other works extend conformal prediction to settings where the standard assumptions may not hold, such as Tibshirani et al. (2019), Prinster et al. (2022), Barber et al. (2023) and Stutz et al. (2023). Our work is based on the cross-conformal prediction method introduced in Vovk (2015) and later extended in Vovk et al. (2018). We refer to Fontana et al. (2023) and Angelopoulos & Bates (2023) for an overview of conformal prediction and its extensions.

The solutions proposed here are based on recent results on the combination of p-values that exploit exchangeability and randomization (Gasparin et al., 2025). The combination of p-values is not new in the statistical literature and dates back at least to Fisher (1948). Fisher's method is based on the assumption of independence among the p-values, an assumption frequently violated in practical applications. Other works propose combination rules valid for arbitrarily dependent p-values; some examples are Rüger (1978), Morgenstern (1980), Rüschendorf (1982), Vovk & Wang (2020), and more recently Vovk et al. (2022b). Clearly, these rules valid under arbitrary dependence come with a price in terms of statistical power. In other words, these methods for combining p-values are usually conservative since they have to protect against the worst-case scenario of dependence. The results in Gasparin et al. (2025) are able to improve these rules valid under arbitrary dependence exploiting the exchangeability of the starting p-values and/or randomization. Their results are derived using extensions of Markov's inequality introduced in Ramdas & Manole (2025).

In the framework of conformal prediction, the combination (or ensembling) of p-values is used in Carlsson et al. (2014), Toccaceli & Gammerman (2017) and Linusson et al. (2017). Their empirical results indicate that Fisher's method is not a valid rule for combining p-values obtained from different splits or algorithms, whereas using rules valid under arbitrary dependence tends to be generally conservative. In particular, Linusson et al. (2017) provides some intuitions suggesting that the empirical coverage of cross-conformal prediction depends on the degree of dependence between the conformal p-values (that it is strictly related to the stability of the underlying prediction algorithm). In a similar spirit, the solutions in Cherubin (2019) and Solari & Djordjilović (2022) aim to combine dependent conformal prediction sets (rather than p-values) derived from different random splits or prediction algorithms. In particular, their approach relies on a majority vote strategy. Gasparin & Ramdas (2024) further extended their approach by introducing a weighting system and incorporating randomization. Other works aim to select or combine conformal prediction sets; see for example Yang & Kuchibhotla (2024) and Liang et al. (2024).

## 3. (Modified) Cross-Conformal Prediction

This section will recap two methods: cross-conformal prediction, and modified cross-conformal prediction. We let $K$ denote the number of folds, and we focus on the (prac-

tical) case when $K$ is small like $K = 5$ or $K = 10$. We will always assume that $m = n/K$ is an integer, which is achievable by only throwing away less than $K$ points from the original dataset. However, we point out in Appendix C that both methods have guarantees without this assumption (which is new to the best of our knowledge, though minor).

### 3.1. Cross-Conformal Prediction

Cross-conformal prediction, introduced by Vovk (2015), is a method to obtain distribution-free prediction intervals. It can be considered as a combination of split conformal prediction (see Appendix A) and cross-validation. It works as follows: data are divided into $K$ disjoint subsets (or folds) $I_1, \ldots, I_K$ of size $m = n/K$. The (cross-validation) scores are defined as:

$$S_i^{\text{CV}} = s\left((X_i, Y_i); \mathcal{D}_{[n] \setminus I_{k(i)}}\right) \quad i = 1, \ldots, n, \quad (2)$$

where $I_{k(i)}$ is the subset containing the $i$-th data point. The cross-conformal prediction set is simply defined as

$$\hat{C}_{n,K,\alpha}^{\text{cross}}(X_{n+1}) = \left\{ y \in \mathcal{Y} : \right.$$

$$\left. \frac{1 + \sum_{i=1}^{n} \mathbb{1}\left\{ s\left((X_{n+1}, y); \mathcal{D}_{[n] \setminus I_{k(i)}}\right) \leq S_i^{\text{CV}} \right\}}{n+1} > \alpha \right\}.$$
$$(3)$$

Vovk et al. (2018) proves that the interval in (3) is such that

$$\mathbb{P}\left(Y_{n+1} \in \hat{C}_{n,K,\alpha}^{\text{cross}}(X_{n+1})\right)$$

$$\geq 1 - 2\alpha - 2(1 - \alpha)\frac{1 - 1/K}{n/K + 1}, \quad (4)$$

where the probability is marginal and is computed with respect to $(X_1, Y_1), \ldots, (X_{n+1}, Y_{n+1})$. In particular, when $K$ is small compared to $n$ (that is, $n \gg K$), the additional term is negligible and the coverage is essentially at least $1 - 2\alpha$. To prove the result in (4), it is useful to define for each subset, $k \in [K]$, the quantity

$$P_k(y) = \frac{1 + \sum_{i \in I_k} \mathbb{1}\left\{ s\left((X_{n+1}, y); \mathcal{D}_{[n] \setminus I_k}\right) \leq S_i^{\text{CV}} \right\}}{m+1},$$
$$(5)$$

that is a discrete p-value if computed using the response test value $Y_{n+1}$ and if data $(X_i, Y_i), i \in [n+1]$, are iid or at least exchangeable (i.e., $\mathbb{P}(P_k(Y_{n+1}) \leq \alpha) \leq \alpha$). This is due to the fact that the scores in $I_k \cup \{n+1\}$ are exchangeable, since the prediction algorithm is trained only on the training points in $[n] \setminus I_k$, so (5) can be seen as a rank-based p-value. It is possible to relate the set defined in

(3) with the cross-conformal p-values in (5). In particular, a point $y$ is included in $\hat{C}_{n,K,\alpha}^{\text{cross}}(X_{n+1})$ if and only if

$$\frac{1}{K} \sum_{k=1}^{K} P_k(y) > \alpha + (1 - \alpha)\frac{K-1}{K+n}. \quad (6)$$

The multiplicative factor of two in the coverage statement in (4) arises from the fact that the average of arbitrarily dependent p-values remains a p-value up to a factor of 2 (Rüschendorf, 1982; Vovk & Wang, 2020):

$$\mathbb{P}\left(\frac{1}{K} \sum_{k=1}^{K} P_k(Y_{n+1}) \leq \alpha\right) \leq 2\alpha. \quad (7)$$

This implies that the statement in (4) can be proved by combining the results in (6) and (7). For a detailed discussion on cross-conformal prediction see, for example, Chapter 4.4 of Vovk et al. (2022a).

*Remark* 3.1. The coverage statement in (4) is meaningless when $K$ is large. In fact, Barber et al. (2021) proves a different bound for the miscoverage rate valid for large $K$. However, in practical applications, the number of splits is usually small if compared with the number of observations (e.g., $K = 5$ or $K = 10$) and the bound in (4) is the one that applies. We discuss the two different bounds and the connection with the CV+ method by Barber et al. (2021) in Appendix B.

*Remark* 3.2. In a regression setting, there are no guarantees that $\hat{C}_{n,K,\alpha}^{\text{cross}}(X_{n+1})$ will be an interval; in fact, there are particular cases where it can be a union of intervals. This property is shared by other "hybrid" methods mentioned in Section 1. One can avoid having a union of intervals by taking the convex hull of the set (the interval formed by the furthest endpoints) as explained in Gupta et al. (2022). In addition, when the residual score is chosen as score function, the prediction set $\hat{C}_{n,K,\alpha}^{\text{cross}}(X_{n+1})$ is a subset of the CV+ set that is guaranteed to be an interval (see Appendix B).

### 3.2. Modified Cross-Conformal Prediction

It is clear from the previous section that we can obtain a set with coverage at least equal to $1 - 2\alpha$ using a modification of the cross-conformal prediction set defined in (3). We define the modified cross-conformal prediction interval (the same name is used in Barber et al. (2021)) as

$$\hat{C}_{n,K,\alpha}^{\text{mod-cross}}(X_{n+1}) = \left\{ y \in \mathcal{Y} : \frac{1}{K} \sum_{k=1}^{K} P_k(y) > \alpha \right\}.$$
$$(8)$$

Using the result stated in (7), we have

$$\mathbb{P}\left(Y_{n+1} \in \hat{C}_{n,K,\alpha}^{\text{mod-cross}}(X_{n+1})\right) \geq 1 - 2\alpha.$$

The intervals defined in (3) and (8) usually have inflated coverage. In other words, with typically employed levels

of $\alpha$, the coverage obtained using these methods often fluctuates between the levels $1 - \alpha$ and 1. This is due to the fact that the rule in (7) is valid under arbitrary dependence and it has to take into account the "worst-case" scenario of dependence, which typically differs from the scenario observed in the data. However, in some situations where the regression algorithm is unstable or with some particular distribution $Q$, the coverage can oscillate between the guaranteed level $1 - 2\alpha$ and $1 - \alpha$. Linusson et al. (2017) offers some empirical observations regarding the miscalibration of the average of p-values obtained from different folds. In particular, since the p-values are dependent, the distribution of the averaged p-values is in between the Bates distribution and the uniform distribution, and this strictly depends on the stability of the underlying algorithm.

Since p-values take discrete values, in order to avoid having noninformative sets identical to $\mathcal{Y}$, the inequality $1 < \alpha(m + 1)$ must hold. A slightly improvement can be obtained using randomized p-values $P_1(Y_{n+1}; \tau), \ldots, P_K(Y_{n+1}; \tau)$ defined by

$$
\begin{aligned}
P_k(y; \tau) &= \\
&= \frac{\tau + \sum_{i \in I_k} \tau \mathbb{1}\left\{ s\left((X_{n+1}, y); \mathcal{D}_{[n] \setminus I_k}\right) = S_i^{\mathrm{CV}} \right\}}{m + 1} + \\
&+ \frac{\sum_{i \in I_k} \mathbb{1}\left\{ s\left((X_{n+1}, y); \mathcal{D}_{[n] \setminus I_k}\right) < S_i^{\mathrm{CV}} \right\}}{m + 1},
\end{aligned}
\tag{9}
$$

where $\tau$ is a uniform random variable in the interval $(0, 1)$ drawn independently from the data. In this case, the p-values (for $y = Y_{n+1}$) are uniformly distributed in the interval $(0, 1)$, rather than taking discrete values. However, the dependence among the p-values obtained from different folds is not broken.

# 4. New Variants of Cross-Conformal Prediction

In this section, we improve the prediction set in (8) using recent results regarding the combination of p-values. In particular, the combination rules that will be used are more powerful than the combinations valid under arbitrary dependence of the p-values. The results are obtained in a completely general manner and do not require the use of expensive computational procedures (Carlsson et al., 2014) or the use of specific models (Boström et al., 2017).

## 4.1. Exchangeable Modified Cross-Conformal Prediction

The interval in (8) can be improved using recent results on the combination of exchangeable p-values. Before proceeding, we state a useful result.

**Proposition 4.1.** *Let* $P_1(Y_{n+1}), \ldots, P_K(Y_{n+1})$ *be the*

*(cross-conformal) p-values obtained using data* $Z_i = (X_i, Y_i)$, $i = [n + 1]$, *then* $P_1(Y_{n+1}), \ldots, P_K(Y_{n+1})$ *are exchangeable, meaning that* $\mathbf{P} \stackrel{d}{=} \mathbf{P}^\pi$, *where* $\stackrel{d}{=}$ *represents equality in distribution,* $\mathbf{P} = (P_1(Y_{n+1}), \ldots, P_K(Y_{n+1}))$, $\mathbf{P}^\pi = (P_{\pi(1)}(Y_{n+1}), \ldots, P_{\pi(K)}(Y_{n+1}))$ *and* $\pi : [K] \to [K]$ *is any permutation of the indices.*

A formal proof of the result is based on the following lemma and is provided in Appendix F.

**Lemma 4.2** (Dean & Verducci (1990); Kuchibhotla (2020))**.** *Suppose* $W = (W_1, \ldots, W_n) \in \mathcal{W}^n$ *is a vector of exchangeable random variables. Fix a transformation* $G : \mathcal{W}^n \to (\mathcal{W}')^m$. *If for each permutation* $\pi_1 : [m] \to [m]$ *there exists a permutation* $\pi_2 : [n] \to [n]$ *such that*

$$
\pi_1 G(w) = G(\pi_2 w), \quad \text{for all } w \in \mathcal{W}^n,
$$

*then* $G(\cdot)$ *preserves exchangeability.*

*Remark* 4.3. The assumption that $n/K = m$ is crucial to prove the result in Proposition 4.1. In fact, if the subsets $I_1, \ldots, I_K$ have different sample sizes, then the result in Proposition 4.1 does not hold. Notice that the p-values in (5) take discrete values $\{1/m, 2/m, \ldots, 1\}$. If the sample sizes differ, then the p-values assume values in different grids of values, and therefore the marginal distributions of $P_1(Y_{n+1}), \ldots, P_K(Y_{n+1})$ are different. This implies that p-values cannot be exchangeable. In addition, with different sample sizes the proof of the result breaks down and a permutation $\pi_2$ that satisfies the condition in Lemma 4.2 does not exist. In Appendix C, we will see how to extend the result to the case where the folds have different sizes using a simple trick.

An improved version of the set in (8) can be defined as:

$$
\hat{C}_{n,K,\alpha}^{\text{e-mod-cross}}(X_{n+1}) = \left\{ y \in \mathcal{Y} : \min_{\ell \in [K]} \frac{1}{\ell} \sum_{k=1}^{\ell} P_k(y) > \alpha \right\},
\tag{10}
$$

where, for a given $y$, the combination of the different $P_k(y)$ is asymmetric and depends on the order of the p-values.

**Theorem 4.4.** *It holds that* $\hat{C}_{n,K,\alpha}^{\text{e-mod-cross}}(X_{n+1}) \subseteq \hat{C}_{n,K,\alpha}^{\text{mod-cross}}(X_{n+1})$. *In addition, if data are exchangeable,*

$$
\mathbb{P}\left( Y_{n+1} \in \hat{C}_{n,K,\alpha}^{\text{e-mod-cross}}(X_{n+1}) \right) \geq 1 - 2\alpha.
\tag{11}
$$

The proof of this and subsequent results is provided in Appendix F.

The theorem indicates that one can derive a set smaller than the modified cross-conformal prediction set while maintaining the same coverage guarantee. The same results hold if the p-values in (9) are used. Specifically, the randomized p-values are still exchangeable if $\tau$ is common across the folds.

Indeed, conditional on $\tau$ the p-values are exchangeable due to Proposition 4.1. In particular, using the p-values in (9) we obtain a smaller set since $P_k(y;\tau) \leq P_k(y)$ almost surely.

*Remark* 4.5. Once $K$ exchangeable (or more generally dependent) p-values are obtained, there are several methods to combine them. The proposed solution is to use the minimum (over $\ell$) of the mean obtained using the first $\ell$ p-values, which is related to the valid combination rule "twice the average" used by Vovk et al. (2018; 2022a) to prove the coverage guarantee of cross-conformal prediction. However, similar results apply to other merging functions like quantiles (for example "twice the median" is also a valid combination rule) and generalized averages (e.g., geometric mean or harmonic mean). However, it is important to emphasize that, as explained in Section 6 of Vovk & Wang (2020), the mean is an effective method of combining strongly dependent p-values, which is often the case for rank-based p-values obtained by cross-conformal prediction. Other merging rules, such as the Bonferroni method, are more appropriate near independence. For instance, Lei et al. (2018, Section 2.3) shows, under some assumptions, that the Bonferroni rule is overly conservative in the context of multisplit conformal prediction.

### 4.2. Randomized Modified Cross-Conformal Prediction

In the previous paragraph, we leveraged the exchangeability of p-values to obtain a smaller set. In this section, we move in a different direction and improve the set (8) using a simple "randomization trick" (introducing a uniform random variable). Indeed, in this case, the exchangeability of the p-values is not necessary. As before, the improvement does not alter the marginal validity of the set, but the new result is obtained in a different way. Although randomization is avoided in some statistical applications due to the extra randomness it introduces, in this case, it does not pose a major issue. Indeed, cross-conformal prediction is, by definition, a randomized method. More precisely, data are randomly divided into $K$ different subsets in the first step, which means that the procedure inherently includes randomness (see Remark 4.8 for further discussion).

We can define a "randomized" improvement of the interval in (8) as follows:

$$
\hat{C}_{n,K,\alpha}^{\text{u-mod-cross}}(X_{n+1}) = \\
\left\{ y \in \mathcal{Y} : \frac{1}{2-U}\frac{1}{K}\sum_{k=1}^{K} P_k(y) > \alpha \right\},
\tag{12}
$$

where $U$ is a uniform random variable in the interval $(0,1)$ independent of all the data.

**Theorem 4.6.** *It holds that* $\hat{C}_{n,K,\alpha}^{\text{u-mod-cross}}(X_{n+1}) \subseteq$

$\hat{C}_{n,K,\alpha}^{\text{mod-cross}}(X_{n+1})$. *In addition, if data are exchangeable,*

$$
\mathbb{P}\left( Y_{n+1} \in \hat{C}_{n,K,\alpha}^{\text{u-mod-cross}}(X_{n+1}) \right) \geq 1-2\alpha.
\tag{13}
$$

Even in this case, the guaranteed marginal coverage remains at least $1-2\alpha$, but the set size is enhanced using a simple result based on randomization.

### 4.3. Exchangeable and Randomized Modified Cross-Conformal Prediction

The results in Section 4.1 and in Section 4.2 can be "combined" in order to obtain a prediction set that improves the one defined in (10). In this case as well, the exchangeability property outlined in Proposition 4.1 is crucial.

We define a randomized improvement of the conformal prediction set defined in (10):

$$
\hat{C}_{n,K,\alpha}^{\text{eu-mod-cross}}(X_{n+1}) = \left\{ y \in \mathcal{Y} : \right. \\
\left. \min\left\{ \frac{1}{2-U}P_1(y), \min_{\ell\in[K]}\frac{1}{\ell}\sum_{k=1}^{\ell} P_k(y) \right\} > \alpha \right\},
\tag{14}
$$

where $U$ is a uniform random variable in the interval $(0,1)$ independent of all the data.

**Theorem 4.7.** *It holds that* $\hat{C}_{n,K,\alpha}^{\text{eu-mod-cross}}(X_{n+1}) \subseteq$ $\hat{C}_{n,K,\alpha}^{\text{e-mod-cross}}(X_{n+1}) \subseteq \hat{C}_{n,K,\alpha}^{\text{mod-cross}}(X_{n+1})$. *In addition, if data are exchangeable,*

$$
\mathbb{P}\left( Y_{n+1} \in \hat{C}_{n,K,\alpha}^{\text{eu-mod-cross}}(X_{n+1}) \right) \geq 1-2\alpha.
\tag{15}
$$

The set in (14) can be considered an improvement of the set described in (10) but not of the (randomized) set in (12), since only the first p-value of the sequence is randomized.

*Remark* 4.8 (Randomization and "interval-hacking"). A direct use of external randomization is present in both procedures described in Section 4.2 and Section 4.3. The use of randomization is often avoided in statistical methods, as it can pose challenges to the reproducibility of results. Clearly, randomization becomes problematic when a human is in the loop and runs the procedure multiple times until the desired result is achieved (for example, in the described cases, one can sample $U$ many times until it reaches a value close to zero). Some recommendations aimed at solving this problem are proposed, for example, in Ramdas & Manole (2025, Section 10). Actually, in the data pipeline of split and cross-conformal prediction methods, randomization comes into play in different parts: by default in the division of data into their respective folds; to smoothen p-values as described in (9); and potentially to improve the conditional coverage as described in Hore & Barber (2024). In particular,

there exists a trade-off between reproducibility and statistical efficiency, and it is not always evident which should be prioritized. In other words, randomized procedures tend to be more efficient than standard procedures but may lack in terms of reproducibility, and vice versa. For instance, our methods may be particularly well-suited in industrial settings, where hundreds or thousands of predictions are made daily, and efficiency may be more important.

### 4.4. Improving Cross-Conformal Prediction

The improvements proposed in the previous subsections are valid for modified cross-conformal prediction; in particular, the new variants are able to produce smaller prediction sets while preserving the same marginal coverage. Specifically, the marginal coverage does not depend on the number of folds $K$ and the number of observations $n$. When the folds have the same size, the techniques can be used to enhance cross-conformal prediction (Vovk, 2015): in particular, by examining (6), one can observe that it is possible to improve cross-conformal prediction simply by replacing the threshold $\alpha$ with $\alpha + (1 - \alpha)(K - 1)/(K + n)$ in the prediction sets defined in (10), (12), and (14).

**Theorem 4.9.** *It holds that*

$$\hat{C}_{n,K,\alpha'}^{\text{e-mod-cross}} \subseteq \hat{C}_{n,K,\alpha}^{\text{cross}},$$

$$\hat{C}_{n,K,\alpha'}^{\text{u-mod-cross}} \subseteq \hat{C}_{n,K,\alpha}^{\text{cross}},$$

$$\hat{C}_{n,K,\alpha'}^{\text{eu-mod-cross}} \subseteq \hat{C}_{n,K,\alpha'}^{\text{e-mod-cross}} \subseteq \hat{C}_{n,K,\alpha}^{\text{cross}},$$

*where $\alpha' = \alpha + (1 - \alpha)(K - 1)/(K + n)$. If data are exchangeable, the marginal coverage of the conformal prediction sets $\hat{C}_{n,K,\alpha'}^{\text{e-mod-cross}}(X_{n+1})$, $\hat{C}_{n,K,\alpha'}^{\text{u-mod-cross}}(X_{n+1})$ and $\hat{C}_{n,K,\alpha'}^{\text{eu-mod-cross}}(X_{n+1})$ is at least $1 - 2\alpha'$.*

In practice, when $n \gg K$, the prediction sets $\hat{C}_{n,K,\alpha}^{\text{cross}}(X_{n+1})$ and $\hat{C}_{n,K,\alpha}^{\text{mod-cross}}(X_{n+1})$ are similar. However, for moderate values of $n$, we will see that the sets defined in (10), (12), and (14) are typically narrower than $\hat{C}_{n,K,\alpha}^{\text{cross}}(X_{n+1})$, even though $\hat{C}_{n,K,\alpha}^{\text{cross}}(X_{n+1})$ assures theoretically a lower coverage guarantee. Clearly, the proposed improvements are valid as long as the marginal coverage level $1 - 2\alpha'$ is meaningful, which in practical applications is the most common case. It follows that the improvements are not valid, for example, in the extreme case of leave-one-out conformal prediction (the case $K = n$).

An experiment using the threshold $\alpha'$ is reported in Appendix E.

## 5. Empirical Results

We study the effectiveness of the proposed methods through a simulation study and real data examples. In all experiments, the score function used is the residual score, defined as:

$$s\left((x, y); \mathcal{D}\right) = |y - \hat{\mu}_{\mathcal{D}}(x)|, \tag{16}$$

where $\hat{\mu}_{\mathcal{D}}$ is the regression function obtained by applying the regression algorithm $\mathcal{A}$ on $\mathcal{D}$.

The code to reproduce the experiments is available at github.com/matteogaspa/EffCrossCP.

### 5.1. Simulation Study

We examine the performance of the proposed methods on simulated data using least squares as our regression algorithm. Data are simulated as in Barber et al. (2021, Section 6); in particular, the number of observations is $n = 100$ and we let the number of regressors vary $p = \{5, 10, \ldots, 200\}$. The training data points are iid from

$$X_i \sim \mathcal{N}_p(0, I_p) \quad \text{and} \quad Y_i \mid X_i \sim \mathcal{N}(X_i^\top \beta, 1),$$

where the vector of coefficients is drawn as $\beta = \sqrt{10}\, v$ for a uniform random unit vector $v \in \mathbb{R}^p$. Ordinary least squares is employed as regression method (if the linear system is underdetermined, then we take the solution that minimizes the $\ell_2$-norm). Formally, given the training data $(X_i, Y_i), i \in [n]$, we estimate the regression function $\hat{\mu}(x) = x^\top \hat{\beta}$, where $\hat{\beta} = X_{\text{mat}}^\dagger Y_{\text{vec}}$, $Y_{\text{vec}}$ is the response vector, $X_{\text{mat}}$ is the matrix of covariates of dimension $n \times p$ and $\dagger$ denotes the Moore-Penrose inverse. The nominal miscoverage rate equals $\alpha = 0.1$, the number of replications (for each $p$) is 1000 and for each replication, we generate a single test point $(X_{n+1}, Y_{n+1})$. The number of folds for cross-conformal prediction and its extensions is $K = 5$.

From Figure 1, we can see a spike in the size observed at $p = 80$. This is due to the fact that the prediction algorithm is unstable when the number of training points is equal (or almost equal) to the number of covariates (Hastie et al., 2022). Since the number of folds equals 5, the peak is observed at $p = 80$.

The smaller size is often observed by the exchangeable and randomized variant of cross-conformal prediction. Cross-conformal prediction (Vovk, 2015), is usually over-conservative, and in some cases, its coverage is closer to one rather than to the guaranteed level. This behavior is not shared by the proposed `e-mod-cross` and `eu-mod-cross`. The coverage of these methods lies between the levels $1 - 2\alpha$ and $1 - \alpha$, and remains essentially constant with respect to the number of covariates $p$. The coverage of the randomized variant `u-mod-cross` depends on $p$ and exhibits a behavior similar to that of standard cross-conformal prediction. In general, our proposals outperform standard cross-conformal prediction in terms of set size.

For additional comparisons, we evaluate our proposed `eu-mod-cross` method with split conformal prediction

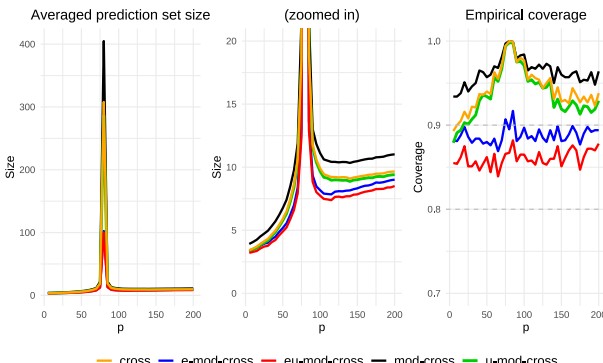 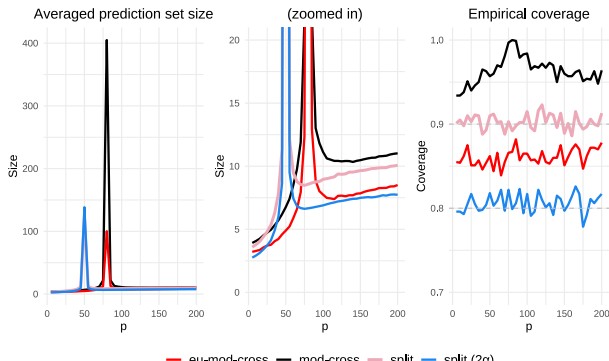

*Figure 1.* Simulation results, showing the size and coverage of the predictive sets for cross-conformal prediction and its variants. In the left plot, peaks are observed at $404, 102, 286, 100$ and $307$ for `mod-cross`, `e-mod-cross`, `u-mod-cross`, `eu-mod-cross` and `cross`, respectively. The parameter $\alpha$ is set to 0.1. The smaller sets are often obtained using `eu-mod-cross` that has coverage between $1 - 2\alpha$ and $1 - \alpha$. The randomized method (`u-mod-cross`) performs similarly to cross-conformal prediction.

*Figure 2.* Simulation results, showing the size and coverage of the predictive intervals obtained using 4 methods. The parameter $\alpha$ is set to 0.1. Split conformal prediction is trained at levels $\alpha$ and $2\alpha$ and it is compared with `mod-cross` and `eu-mod-cross`. The modified cross-conformal prediction method always overcovers and tends to produce large prediction sets. Its exchangeable and randomized variant gives good results in terms of size. When $p \in [25, 60]$, the average size of the `eu-mod-cross` method is smaller than that of the split conformal prediction method trained at level $2\alpha$.

trained at levels $\alpha$ and $2\alpha$. In particular, we note that the marginal coverage of the exchangeable and randomized variant is at least $1 - 2\alpha$. From Figure 2, we can observe that for some values of $p \in [25, 60]$, when the prediction algorithm is not stable for the split conformal prediction method, the average length of the `eu-mod-cross` sets is smaller than that of split conformal prediction trained at level $2\alpha$. Described differently, both techniques ensure the same coverage level. However, there is no single method that performs best for all values of $p$. When $p$ is sufficiently small compared to $n$ and the algorithm is stable, split conformal prediction trained at level $\alpha$ performs well, although it uses half the points to train the model.

Additional results, comparing the proposed variants with other conformal prediction methods, such as jackknife+ and full conformal prediction, are reported in Appendix D.

## 5.2. Real Data Application

We apply the proposed methods to the "Online News Popularity" dataset (Fernandes et al., 2015). The dataset contains information on $n = 39\,797$ articles published by the online news blog Mashable. After some preprocessing operations, the number of covariates is $p = 55$ and the covariates contain information about the text of the article. The goal is to predict the number of times the article was shared on a logarithmic scale. Three different regression algorithms are used, specifically: linear regression (as described in Section 5.1), lasso regression with penalty parameter set to 0.2 and random forest with 200 trees grown for each forest.

Conformal prediction methods are applied to $10\,000$ data points randomly sampled without replacement; while other $2500$ observations chosen at random from those not part of the training set are used as the test set. The miscoverage rate is set to $\alpha = 0.1$ and the procedure is repeated 100 times to remove the randomness of the split. The method used are cross-conformal prediction and its variants (with $K = 10$) and split conformal prediction. In particular, split conformal prediction is trained both at levels $\alpha$ and $2\alpha$. The averages over 100 trials are reported as results in Figure 3 and Table 1.

From Figure 3, it is possible to note that cross-conformal prediction and its modified version give very similar results in terms of size and are usually slightly better than split conformal prediction trained at level $\alpha$. The randomized methods `u-mod-cross` and `eu-mod-cross` show a significant improvement in terms of size. The improvement is not as evident for the `e-mod-cross` method, which turns out to be slightly better than the modified method. The smaller sets are obtained using split conformal prediction trained at level $2\alpha$. The level of coverage of cross-conformal prediction (and `mod-cross`) is around $1 - \alpha$ and the two methods tend to overcover (indeed, they guarantee a miscoverage rate smaller than $2\alpha$). The `e-mod-cross` method exhibits similar performance to cross-conformal prediction in terms of coverage; while the coverage of `u-mod-cross` and `eu-mod-cross` is between the levels $1 - 2\alpha$ and $1 - \alpha$. Clearly, there exists a direct relationship between

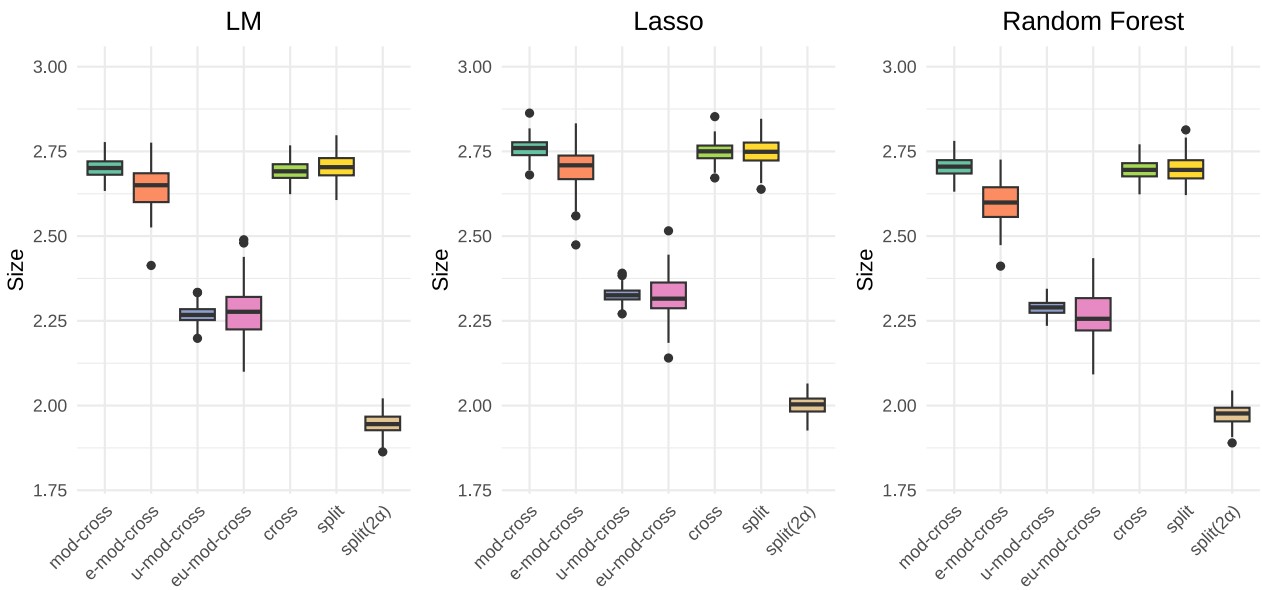

*Figure 3.* Empirical size obtained using different regression algorithms and different conformal prediction methods. The methods `mod-cross` and `cross` give similar results. The variants that use randomization (`u-mod-cross` and `eu-mod-cross`) have a smaller size with respect to the other methods trained at level $\alpha$. The smaller sets are obtained using split conformal prediction trained at level $2\alpha$.

*Table 1.* Empirical coverage for the News Popularity Dataset using different regression algorithms and different conformal prediction methods. The $\alpha$-level is set to 0.1. `Mod-cross` and `cross` have empirical coverage around $1 - \alpha$ (while guaranteeing a coverage level of at least $1 - 2\alpha$). The coverage for the randomized methods lies between $1 - 2\alpha$ and $1 - \alpha$. The `e-mod-cross` variant has coverage $\approx 1 - \alpha$. Coverage variability is reported through the minimum and maximum values across the 100 replications, together with their difference.

| Method | Metric | mod-cross | e-mod-cross | u-mod-cross | eu-mod-cross | cross | split | split($2\alpha$) |
|--------|--------|-----------|-------------|-------------|--------------|-------|-------|------------------|
|      | Mean  | 0.902 | 0.896 | 0.851 | 0.852 | 0.901 | 0.900 | 0.801 |
| LM   | Min   | 0.884 | 0.876 | 0.834 | 0.826 | 0.884 | 0.885 | 0.777 |
|      | Max   | 0.917 | 0.915 | 0.874 | 0.881 | 0.916 | 0.916 | 0.831 |
|      | Range | 0.032 | 0.039 | 0.040 | 0.055 | 0.032 | 0.030 | 0.054 |
|      | Mean  | 0.901 | 0.896 | 0.851 | 0.845 | 0.900 | 0.900 | 0.800 |
| Lasso| Min   | 0.888 | 0.872 | 0.831 | 0.826 | 0.887 | 0.884 | 0.772 |
|      | Max   | 0.916 | 0.914 | 0.870 | 0.876 | 0.915 | 0.915 | 0.822 |
|      | Range | 0.028 | 0.042 | 0.039 | 0.050 | 0.028 | 0.030 | 0.050 |
|      | Mean  | 0.903 | 0.892 | 0.853 | 0.847 | 0.903 | 0.900 | 0.801 |
| RF   | Min   | 0.885 | 0.870 | 0.832 | 0.815 | 0.885 | 0.882 | 0.778 |
|      | Max   | 0.918 | 0.915 | 0.875 | 0.881 | 0.917 | 0.918 | 0.819 |
|      | Range | 0.032 | 0.045 | 0.042 | 0.066 | 0.032 | 0.036 | 0.042 |

set size and coverage, with smaller prediction sets attaining coverage values closer to $1 - 2\alpha$. The coverage of split conformal prediction is essentially equal to the target levels $1 - \alpha$ or $1 - 2\alpha$ (see Appendix A for further details on the coverage properties of the method). As a final remark, the `eu-mod-cross` method demonstrates a higher degree of variability in coverage, while the remaining methods exhibit

comparable and more stable variability levels.

Additional experiments on real-world datasets are reported in Appendix E.

*Table 2.* Comparison of the properties of different conformal prediction methods. The first two columns regards the marginal coverage. The theoretical guarantees are valid in finite sample. The last column counts the number of times that the algorithm $\mathcal{A}$ is run on a data set containing $n$ training points, for obtaining a prediction set for a new test point. The parameter $n_{\mathrm{grid}}$ represents the number of different possible $y$ values.

| Method | Theoretical guarantee | Typical empirical coverage | Model training cost |
|---|---|---|---|
| Split | $\geq 1 - \alpha$ | $\approx 1 - \alpha$ | 1 |
| Full | $\geq 1 - \alpha$ | $\approx 1 - \alpha$ or $> 1 - \alpha$ if $\hat{\mu}$ overfits | $n_{\mathrm{grid}}$ |
| Jackknife+ | $\geq 1 - 2\alpha$ | $\approx 1 - \alpha$ | $n$ |
| Cross | $\geq 1 - 2\alpha - 2/\sqrt{n}$ | $\geq 1 - \alpha$ | $K$ |
| Mod-cross | $\geq 1 - 2\alpha$ | $> 1 - \alpha$ | $K$ |
| e/u/eu-mod-cross | $\geq 1 - 2\alpha$ | $\in [1 - 2\alpha, 1 - \alpha]$ | $K$ |

# 6. Discussion

We present new variants of cross-conformal prediction that can achieve smaller prediction sets while maintaining valid coverage guarantees. The achievements are based on recent results on the combination of dependent and exchangeable p-values (Gasparin et al., 2025). In particular, starting from a target miscoverage rate equal to $\alpha$, the new methods guarantee a marginal coverage of at least $1 - 2\alpha$. The same coverage is guaranteed by other methods, such as the jackknife+ introduced by Barber et al. (2021) or multi-split conformal prediction (with threshold set to a half) by Solari & Djordjilović (2022).

Specifically, similar to cross-validation, the proposed approaches require training the models only $K$ times, unlike $n$ times for the jackknife+ or even potentially an infinite number of times for full conformal prediction (see Table 2). The empirical coverage of the proposed methods usually oscillates between levels $1 - 2\alpha$ and $1 - \alpha$, while for standard cross-conformal prediction the empirical coverage is usually around the nominal $1 - \alpha$ level. As reported in the experimental results, the size of the sets is smaller than that obtained by split conformal prediction and cross-conformal prediction. Since the results depend on randomized or asymmetric combinations of p-values, the size is generally, though not consistently, more variable compared to cross-conformal prediction. In particular, while randomization and asymmetric combination improve efficiency of the prediction sets, they add an extra-layer of randomness to the procedure. The results presented in Sections 4.1, 4.2 and 4.3 can also be extended to the case where the number of observations in the folds are different. Cross-conformal prediction can be improved using the same techniques, as shown in Section 4.4.

The question of which variant of cross-conformal prediction to use in practice is a subtle one. If it is crucial to obtain a $1 - \alpha$ guarantee against worst-case distributions and unstable algorithms (for example, when downstream decisions critically depend on the provided theoretical guarantees), then it makes sense to run our methods at level $\alpha/2$. On the other hand, if conformal prediction is used merely as "weak guidance" for downstream decisions, meaning the user is willing to tolerate violations of the $1 - \alpha$ target, then we recommend running the proposed cross-conformal variants at level $\alpha$ (where the worst-case coverage is $1 - 2\alpha$, but actual coverage lies between $1 - 2\alpha$ and $1 - \alpha$). Finally, if the situation is intermediate, where conformal prediction is neither applied extremely rigorously nor used as loose guidance, and the goal is to achieve $1 - \alpha$ coverage for "typical" datasets and algorithms, while accepting both overcoverage and undercoverage for unusual distributions or algorithms, then (modified) cross-conformal prediction may be the most appropriate choice.

In general, the sets presented in Section 4 show good properties in terms of size and coverage, both in simulations and in applications. The methods can be especially useful in settings where full conformal prediction or jackknife+ are computationally prohibitive.

## Impact Statement

This paper presents work whose goal is to advance the field of Machine Learning. There are some potential societal consequences of our work, some of these have been carefully considered and are explicitly outlined and emphasized within the text of the paper.

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

## A. Split Conformal Prediction

We briefly describe the split (or inductive) conformal prediction method introduced in Papadopoulos et al. (2002); Lei et al. (2018). We assume that we are in the same setup described in Section 2 and the goal is to obtain a prediction set for the response value $Y_{n+1}$ given the training data and covariates in $X_{n+1}$. In this case, data are divided into two disjoint subsets $\mathcal{D}_{\text{train}}$ and $\mathcal{D}_{\text{cal}}$. The algorithm is trained using the data points in $\mathcal{D}_{\text{train}}$, while the scores $S_i := s\left((X_i, Y_i); \mathcal{D}_{\text{train}}\right), i \in \mathcal{D}_{\text{cal}}$, are obtained from the observations in the calibration set. The split conformal prediction set is simply defined as

$$\hat{C}_{n,\alpha}^{\text{split}}(X_{n+1}) = \left\{ y \in \mathcal{Y} : s\left((X_{n+1}, y); \mathcal{D}_{\text{train}}\right) \leq \hat{q} \right\}, \tag{17}$$

where $\hat{q} := \text{quantile}\left(S_1, \ldots, S_{|\mathcal{D}_{\text{cal}}|}; (1-\alpha)(1 + 1/|\mathcal{D}_{\text{cal}}|)\right).$[1] The set in (17) can be re-written as

$$\hat{C}_{n,\alpha}^{\text{split}}(X_{n+1}) = \left\{ y \in \mathcal{Y} : P(y) > \alpha \right\},$$

where

$$P(y) = \frac{1 + \sum_{i \in \mathcal{D}_{\text{cal}}} \mathbb{1}\left\{ s\left((X_{n+1}, y); \mathcal{D}_{\text{train}}\right) \leq s\left((X_i, Y_i); \mathcal{D}_{\text{train}}\right) \right\}}{|\mathcal{D}_{\text{cal}}| + 1},$$

that is a p-value when calculated in $Y_{n+1}$ similar to the one defined in (5). This implies that, if the data are iid, the marginal coverage of the set $\hat{C}_{n,\alpha}^{\text{split}}(X_{n+1})$ is at least $1 - \alpha$. In addition, if the residuals have no ties (they have a continuous joint distribution) then

$$\mathbb{P}\left(Y_{n+1} \in \hat{C}_{n,\alpha}^{\text{split}}(X_{n+1})\right) \leq 1 - \alpha + \frac{1}{|\mathcal{D}_{\text{cal}}| + 1}.$$

The proof of the result can be found in Lei et al. (2018), and the result states that the marginal coverage is essentially $1 - \alpha$ when the number of observations is sufficiently large.

One of the attractive properties of split conformal prediction is that the computational cost of the procedure is low compared to that of full (or transductive) conformal prediction. In fact, the model only needs to be trained once, and the predictions are then calibrated using the data points in $\mathcal{D}_{\text{cal}}$.

## B. Marginal Coverage of Cross-Conformal Prediction and Connection with CV+

As stated in Remark 3.1, it is possible to establish an alternative bound for the marginal coverage of cross-conformal prediction, distinct from the one shown in (4). In particular, Barber et al. (2021) proves that

$$\mathbb{P}\left(Y_{n+1} \in \hat{C}_{n,K,\alpha}^{\text{cross}}(X_{n+1})\right) \geq 1 - 2\alpha - 2(1-\alpha)\frac{1 - K/n}{K+1}. \tag{18}$$

The proof technique is completely different from the technique presented in Section 3 based on p-values and relies on counting arguments applied to tournament matrices (Barber et al., 2021; Angelopoulos et al., 2024). Combining the results in (4) and (18), we obtain

$$\mathbb{P}\left(Y_{n+1} \in \hat{C}_{n,K,\alpha}^{\text{cross}}(X_{n+1})\right) \geq 1 - 2\alpha - 2(1-\alpha)\min\left\{\frac{1 - 1/K}{n/K + 1}, \frac{1 - K/n}{K+1}\right\} \geq 1 - 2\alpha - 2/\sqrt{n}. \tag{19}$$

The two bounds are compared in Figure 4 and it is possible to see that the two bounds have opposite behaviors. As depicted in Figure 4, even for small or moderate $n$, the bound in (4) is the one that applies to commonly employed values of $K$.

In addition, in a regression setting, cross-conformal prediction (Vovk, 2015) is closely related to $K$-fold CV+ introduced in Barber et al. (2021). In particular, both methods can be used to obtain prediction sets with finite sample coverage guarantees. Cross-conformal prediction is covered in Section 3; here, we introduce CV+ and explain its connection to cross-conformal prediction. In this case as well, the data points are divided into $K$ disjoint folds $I_1, \ldots, I_K$ of size $m = n/K$, and $\hat{\mu}_{-I_k}$ refers to the regression function trained using data in $[n] \setminus I_k, k \in [K]$. The $K$-fold CV+ prediction set is defined as

$$\hat{C}_{n,K,\alpha}^{\text{CV+}}(X_{n+1}) = \Bigg[ -\text{quantile}\left(\left(-\left(\hat{\mu}_{-I_{k(i)}}(X_{n+1}) - S_i^{\text{CV+}}\right)\right)_{i \in [n]}; (1-\alpha)(1 + 1/n)\right),$$

$$\text{quantile}\left(\left(\hat{\mu}_{-I_{k(i)}}(X_{n+1}) + S_i^{\text{CV+}}\right)_{i \in [n]}; (1-\alpha)(1 + 1/n)\right)\Bigg], \tag{20}$$

---

[1]We define $\text{quantile}(z; \gamma) = \inf\{a : n^{-1}\sum_{i=1}^{n} \mathbb{1}\{z_i \leq a\} \geq \gamma\}$, for any $z \in \mathbb{R}^n$.

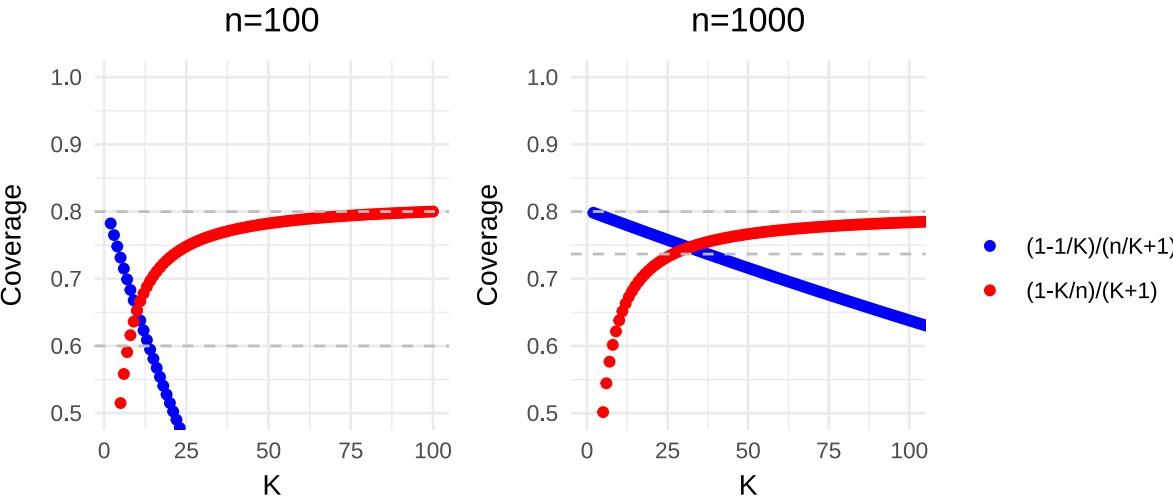

*Figure 4.* Comparison of the bounds in (4) and (18) for different values of $K$ with $\alpha = 0.1$. Dashed lines represent the levels $1 - 2\alpha - 2/\sqrt{n}$ and $1 - 2\alpha$.

where $S_i^{\mathrm{CV+}} = |Y_i - \hat{\mu}_{-I_{k(i)}}(X_i)|, i \in [n]$, are the residual scores (or absolute residuals). An attractive property of the set in (20) is that it is interpretable, since it is always an interval (rather than, possibly, a union of intervals). In particular, when $n = K$, it corresponds to the jackknife+ interval by Barber et al. (2021).

At first glance, the sets $\hat{C}_{n,K,\alpha}^{\mathrm{cross}}(X_{n+1})$ and $\hat{C}_{n,K,\alpha}^{\mathrm{CV+}}(X_{n+1})$ can appear distinct; however, Barber et al. (2021, Appendix B.2) proves that, when the score function in (3) is the residual score, then

$$\hat{C}_{n,K,\alpha}^{\mathrm{cross}}(X_{n+1}) \subseteq \hat{C}_{n,K,\alpha}^{\mathrm{CV+}}(X_{n+1}).$$

As a corollary, it follows that the marginal coverage guarantee in (19) also holds for the CV+ method. This implies that the marginal coverage guarantee for the jackknife+, the case $K = n$, is at least $1 - 2\alpha$. The empirical coverage often exceeds the stated $1 - 2\alpha - 2/\sqrt{n}$, typically aligning closer to the level $1 - \alpha$, and sometimes approaching one. In fact, the value $2\alpha$ can be considered as the worst-case scenario for the method. Under some assumptions about the stability of the prediction algorithm, a modified version of the jackknife+ is shown to have marginal coverage close to $1 - \alpha$. A similar problem is studied in Steinberger & Leeb (2023), where the authors prove a conditional coverage probability statement for $K$-fold cross-validation (a set similar to the one in (20)) valid under some assumptions on the algorithm and the distribution of the data. We refer to Angelopoulos et al. (2024, Ch. 6) for a detailed discussion of the properties of cross-conformal prediction and jackknife+.

## C. Cross-Conformal Prediction with Varying Fold Sizes

In this Section we treat the case where the number of observation in each fold can differ. We consider the same setup as at the beginning of Section 3, and we allow different sizes among the subsets. Let $m_k$ denote the number of observations in subset $I_k, k \in [K]$. By definition, the sum $m_1 + \cdots + m_K$ equals the number of observations $n$. In this case, the definition of conformal p-values in (5) change slightly, allowing for dependence on $m_k$ in the denominator:

$$P_k(y) = \frac{1 + \sum_{i \in I_k} \mathbb{1}\left\{s\left((X_{n+1}, y); \mathcal{D}_{[n] \setminus I_k}\right) \leq S_i^{\mathrm{CV}}\right\}}{m_k + 1}, \tag{21}$$

where $S_i^{\mathrm{CV}}$ is defined in (2). However, $\mathbb{P}(P_k(Y_{n+1}) \leq \alpha) \leq \alpha$, still holds for any $\alpha \in (0, 1)$. In addition, we define the weights

$$w_k = \frac{m_k + 1}{n + K}, \tag{22}$$

where we note that the weights are positive, sum to one and it holds that $w_k = 1/K$ if $m_1 = \cdots = m_K$.

It is now possible to prove that the marginal coverage of cross-conformal prediction with varying fold sizes remains the same.

**Lemma C.1.** *Suppose that $m_k = |I_k|, k \in [K]$, then the set $\hat{C}_{n,K,\alpha}^{\text{cross}}(X_{n+1})$ in (3), is such that*

$$\mathbb{P}\big(Y_{n+1} \in \hat{C}_{n,K,\alpha}^{\text{cross}}(X_{n+1})\big) \geq 1 - 2\alpha - 2(1-\alpha)\frac{1 - 1/K}{n/K + 1}.$$

*Proof.* According to the definition of the cross-conformal prediction set in (3), we can see that $y \in \hat{C}_{n,K,\alpha}^{\text{cross}}(X_{n+1})$ if and only if

$$\frac{1 + \sum_{i=1}^{n} \mathbb{1}\left\{s\left((X_{n+1}, y); \mathcal{D}_{[n] \setminus I_{k(i)}}\right) \leq S_i^{\text{CV}}\right\}}{n + 1} > \alpha \iff \sum_{k=1}^{K} w_k P_k(y) > \alpha + (1-\alpha)\frac{K-1}{n+K}, \tag{23}$$

where $w_k$ and $P_k(y)$ are defined in (22) and (21), respectively. To complete the proof, we apply the fact that the weighted average of p-values provides a quantity that is a p-value up to a factor of 2 (Vovk & Wang, 2020). $\square$

At this point, one may wonder whether the validity of the sets defined in Sections 3.2, 4.1, 4.2 and 4.3 can also be extended to the case where the fold sizes vary. Since twice the (simple) average of p-values is itself a p-value under arbitrary dependence of the starting p-values, it follows that the coverage guarantee of sets $\hat{C}_{n,K,\alpha}^{\text{mod-cross}}(X_{n+1})$ and $\hat{C}_{n,K,\alpha}^{\text{u-mod-cross}}(X_{n+1})$ is preserved even if $P_1(y), \ldots, P_K(y)$ are obtained using different $m_k$.

The coverage guarantee for sets $\hat{C}_{n,K,\alpha}^{\text{e-mod-cross}}(X_{n+1})$ and $\hat{C}_{n,K,\alpha}^{\text{eu-mod-cross}}(X_{n+1})$ is valid when the underlying p-values are exchangeable, and this is related to the number of data points in each fold, as stated in Remark 4.3. However, p-values can be made exchangeable through a random permutation of the indices. For example, assume $n = 101$ and $K = 5$; in this case, the subset with 21 observations does not always have to be the same, but should be randomly selected among the 5 folds. This implies that the coverage guarantees for the sets $\hat{C}_{n,K,\alpha}^{\text{e-mod-cross}}(X_{n+1})$ and $\hat{C}_{n,K,\alpha}^{\text{eu-mod-cross}}(X_{n+1})$ can still hold.

More attention should be paid to Section 4.4. In fact, as seen on the right side of (23), $y$ belongs to the set only if the weighted average of the conformal p-values exceeds a certain threshold. However, the weighted average is an asymmetric function, and results on the combination of exchangeable p-values do not hold in this case. Only the result using randomization remains valid.

## D. Additional Results Related to Section 5.1

We compare the results obtained in Section 5.1 with split conformal, full conformal prediction, and jackknife+. In addition, `eu-mod-cross` conformal prediction is added for comparison. Full and split conformal prediction are fitted using the package R **conformalInference**. The simulation scenario considered is the same as that described in Section 5.1 and all methods are trained at level $\alpha = 0.1$. The theoretical and empirical guarantees and the computational cost of the methods are reported in Table 2 (Section 6). The different methods will be compared in terms of coverage and interval size.

Also in Figure 5, we can see some spikes in the width of the sets at different levels of $p$. Since split conformal prediction uses $n/2$ data points to train the model, the peak is observed at $p = 50$; while for the jackknife+ this peak is observed at $p = 100$. However, the peak for the jackknife+ is smaller than that observed for split conformal prediction and the `eu-mod-cross` method. The smaller sets are usually obtained using `eu-mod-cross` conformal prediction (or jackknife+). It is important to note that the jackknife+ has the same coverage guarantee as `eu-mod-cross` conformal prediction; however, the empirical coverage for the jackknife+ is around the level $1 - \alpha$ while for our method it lies between levels $1 - 2\alpha$ and $1 - \alpha$. As reported in Table 2, the computational cost of the jackknife+ is higher than that of cross-conformal prediction methods: it requires $n$ calls to the prediction algorithm (versus the $K$ required by cross-conformal methods). However, the method is non-randomized, since it can be seen as an extension of cross-conformal prediction to the extreme case $K = n$.

As observed in Barber et al. (2021), when $p > n$, full conformal prediction results in intervals of infinite length because for each possible value of the response, all residuals are equal to zero. In practice, the interval is truncated to a finite range, which has a minimal effect on the marginal coverage (Chen et al., 2018). Split conformal prediction and jackknife+ have similar coverage, but the intervals obtained from jackknife+ are usually smaller (except when the algorithm proves to be unstable).

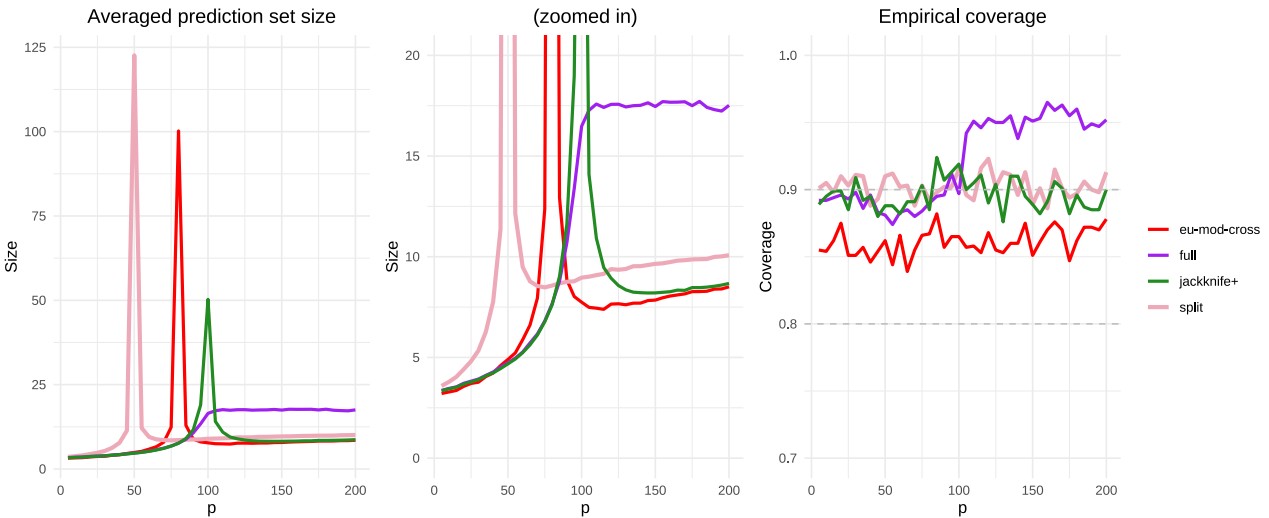

*Figure 5.* Simulation results, showing the size and the coverage of the predictive intervals for jackknife+, split conformal prediction and full conformal prediction. The `eu-mod-cross` method is added for comparison and the $\alpha$-level is set to $0.1$. The smaller sets are usually observed by `eu-mod-cross` conformal prediction. Split conformal prediction and jackknife+ have empirical coverage $\approx 1 - \alpha$.

## E. Additional Experiments

**Communities and Crime dataset.** We apply the proposed methods to the Communities and Crime dataset (Redmond, 2002). The dataset contains information on $n = 1994$ communities in the United States and the goal is to predict the per capita violent rate. After removing the columns containing missing values and categorical variables, the number of regressors is $p = 99$. Two regression algorithms are used, specifically lasso regression with penalty parameter set to $0.01$ and random forest with 50 trees grown for each forest.

The $\alpha$-level is set to $0.1$ and the conformal prediction methods are applied on 1000 data points randomly sampled without replacement. The remaining part is used as a test set to compute the metrics. The procedure is repeated 100 times to remove the randomness of the split and we report the averages over these 100 trials. The methods used are cross-conformal prediction and its variants (with $K = 10$), and split conformal prediction is added for comparison.

The results are reported in Figure 6, where it is possible to see that the smaller sets are obtained using the `eu-mod-cross` method. The modified variants using exchangeability and randomization exhibit higher variability in interval width, likely due to the use of randomization and the asymmetry of the combination rules. All proposed methods have an empirical coverage of at least $1 - 2\alpha$. We remark that cross-conformal prediction guarantees a coverage of at least $1 - 2\alpha$, but is usually conservative. The coverage of the new methods is closer to the level $1 - 2\alpha$ and the new variants outperform cross-conformal prediction in terms of set size.

**Boston Housing dataset.** We apply conformal prediction methods on a dataset of moderate dimensions, with $p = 13$ and $n = 506$. The aim is to predict the cost of a house in Boston given some information on the neighborhood. The algorithm used is standard linear regression. We apply conformal prediction methods using 200 training points, the remaining part is used as test set. The number of different subsets for cross-conformal prediction is set to $K = 5$ and the miscoverage rate is $\alpha = 0.1$. The procedure is repeated 100 times, and we report the averages over the 100 replications.

From Table 3, we see that smaller sets are obtained using `eu-mod-cross` conformal prediction, while larger ones are produced by split conformal prediction, which exhibits high variability in set size. The methods `e-mod-cross` and `u-mod-cross` have an empirical coverage slightly lower than $1 - \alpha$, with an average size generally smaller than that obtained using cross-conformal prediction. Full conformal prediction exhibits low variability in terms of size, with the sets typically being smaller than those produced by split conformal prediction and cross-conformal prediction. However, as already seen, these advantages are counterbalanced by a high computational cost.

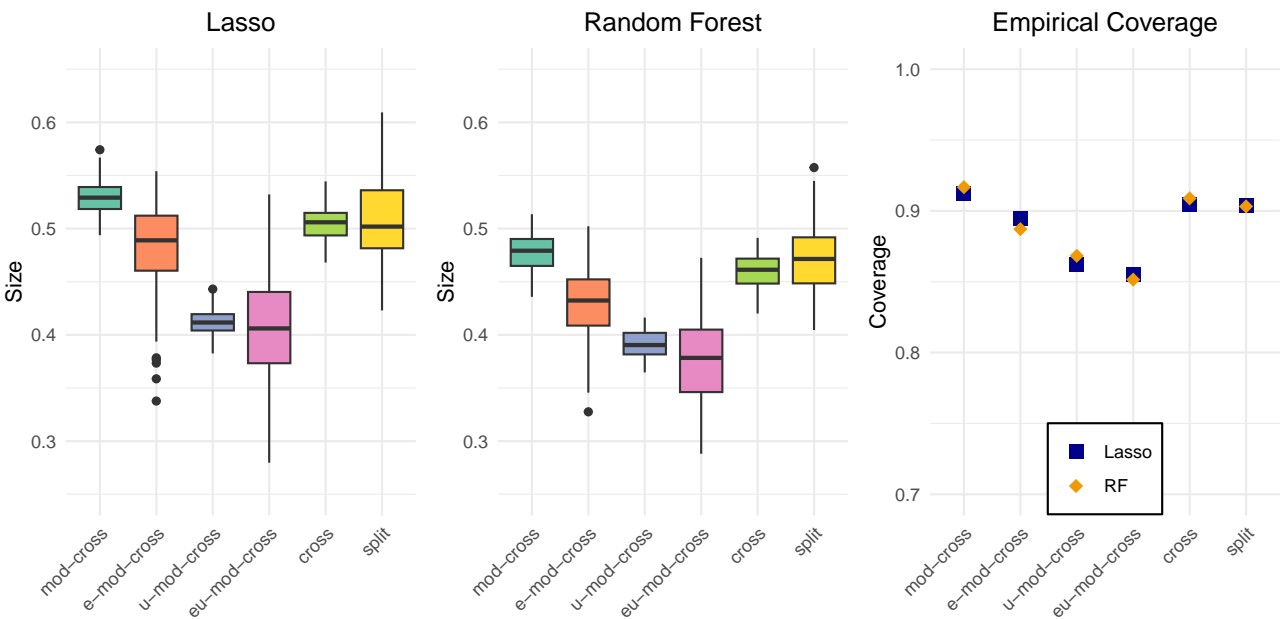

*Figure 6.* Results for the "Communities and Crime" dataset. Empirical size and empirical coverage of different conformal prediction algorithms are reported. The $\alpha$-level is set to $0.1$. The smaller sets are obtained using `eu-mod-cross` whose empirical coverage is around $0.85$. Cross-conformal prediction is conservative, but it tends to produce stable sets.

*Table 3.* Results for the "Boston Housing" dataset using OLS as regression algorithm. The results refer to set size, except for the last row, which refers to the marginal coverage. The $\alpha$-level is set to $0.1$. The smaller sets are obtained using `eu-mod-cross` conformal prediction. The variability is especially high when using split conformal prediction.

|          | mod-cross | e-mod-cross | u-mod-cross | eu-mod-cross | cross  | split  | full   |
| -------- | --------- | ----------- | ----------- | ------------ | ------ | ------ | ------ |
| Mean     | 17.303    | 14.920      | 14.143      | 13.370       | 15.753 | 16.357 | 14.516 |
| Sd       | 1.440     | 1.998       | 1.152       | 1.899        | 1.307  | 2.751  | 1.244  |
| Median   | 17.188    | 14.692      | 14.113      | 13.128       | 15.679 | 16.041 | 14.656 |
| Min      | 13.883    | 10.068      | 11.551      | 8.822        | 12.766 | 11.730 | 11.998 |
| Max      | 20.935    | 20.537      | 17.150      | 18.339       | 19.101 | 27.469 | 17.756 |
| Coverage | 0.927     | 0.888       | 0.878       | 0.855        | 0.908  | 0.897  | 0.888  |

**UPDRS dataset.** We tested our methods on a dataset containing information on patients with early-stage Parkinson's disease (Tsanas & Little, 2009). The goal is to predict the total UPDRS (Unified Parkinson's Disease Rating Scale) using a range of biomedical voice measurements. In particular, after some preprocessing operations, the data set includes $n = 5875$ points and $p = 13$ covariates. The two regression algorithms used are lasso regression (with penalty parameter equal to $0.01$) and random forest (with 25 trees grown for each forest). The $\alpha$-level is set to $0.1$, the number of folds is $K = 10$ and the conformal prediction methods are applied on 3000 data points randomly sampled without replacement. The remaining part is used as a test set to compute the metrics. The procedure is repeated 100 times to remove the randomness of the split. The results reported are the averages over these 100 trials. We compare our proposals with cross-conformal prediction. In addition, split conformal prediction with miscoverage rate set to $\alpha$ and $2\alpha$ is added for comparison.

The results are reported in Table 4 and Table 5. In Table 4, we can see that for lasso regression the smaller sets are obtained using split conformal with miscoverage rate set to $2\alpha$. Overall, our approaches typically yield smaller sets compared to those obtained using standard cross-conformal prediction and split conformal prediction. However, we can observe a higher variability derived from the use of randomization (or sequential processing of the p-values). Interestingly, when the random forest is used as a regression algorithm (Table 5), the smaller sets are obtained using the exchangeable and randomized cross-conformal prediction method. In particular, on average, the method also outperforms the split conformal prediction

trained at level $2\alpha$. In both cases, the marginal coverage of the proposed methods fluctuates between levels $1 - 2\alpha$ and $1 - \alpha$. On the other hand, from Table 5 it is possible to see that cross conformal is conservative with coverage higher than the nominal $1 - \alpha$.

*Table 4.* Results for the UPDRS dataset using lasso as regression algorithm. The results refer to set size, except for the last row, which refers to the marginal coverage. The $\alpha$-level is set to 0.1. On average the smaller sets are obtained using split conformal with miscoverage rate $2\alpha$. The marginal coverage of the proposed methods lies between $1 - 2\alpha$ and $1 - \alpha$.

|          | mod-cross | e-mod-cross | u-mod-cross | eu-mod-cross | cross  | split  | split $(2\alpha)$ |
|----------|-----------|-------------|-------------|--------------|--------|--------|-------------------|
| Mean     | 29.965    | 28.956      | 26.427      | 26.252       | 29.686 | 29.901 | 23.773            |
| Sd       | 0.404     | 1.019       | 1.744       | 1.942        | 0.383  | 0.878  | 0.402             |
| Median   | 29.946    | 29.175      | 26.202      | 26.092       | 29.725 | 29.703 | 23.734            |
| Min      | 11.450    | 10.019      | 8.477       | 9.248        | 11.340 | 28.539 | 22.913            |
| Max      | 32.808    | 30.826      | 31.267      | 30.716       | 32.698 | 32.318 | 25.223            |
| Coverage | 0.904     | 0.892       | 0.854       | 0.850        | 0.901  | 0.901  | 0.800             |

*Table 5.* Results for the UPDRS dataset using random forest as regression algorithm. The results refer to set size, except for the last row, which refers to the marginal coverage. The $\alpha$-level is set to 0.1. On average the smaller sets are obtained using the exchangeable and randomized version of cross-conformal prediction. The marginal coverage of the proposed methods lies between $1 - 2\alpha$ and $1 - \alpha$.

|          | mod-cross | e-mod-cross | u-mod-cross | eu-mod-cross | cross  | split  | split $(2\alpha)$ |
|----------|-----------|-------------|-------------|--------------|--------|--------|-------------------|
| Mean     | 17.180    | 15.186      | 14.737      | 13.718       | 17.015 | 19.210 | 14.550            |
| Sd       | 0.918     | 1.058       | 1.481       | 1.370        | 0.908  | 0.825  | 0.638             |
| Median   | 17.175    | 15.303      | 14.643      | 13.762       | 16.954 | 19.232 | 14.541            |
| Min      | 8.697     | 7.376       | 6.716       | 5.725        | 8.697  | 16.842 | 13.079            |
| Max      | 23.890    | 17.945      | 22.129      | 17.725       | 23.560 | 20.954 | 16.551            |
| Coverage | 0.931     | 0.883       | 0.887       | 0.846        | 0.929  | 0.900  | 0.802             |

**Abalone dataset.** The proposed methods are applied to the abalone dataset (Nash et al., 1994). The goal is to predict the age of abalones (the number of rings) using $p = 8$ physical measurements. The dataset contains $n = 4177$ observations where 4000 observations are used as training points, while the remaining part is used as a test set. In this experiment, we directly modify the cross-conformal prediction as described in Section 4.4 (indeed, we remove the word `mod` from the labels in Tables 6, 7 and 8). The procedure is repeated 100 times to remove the randomness of the split and the results reported are the average over the 100 trials. The $\alpha$-level is set to 0.1 and we use different number of folds, in particular, $K = \{5, 10, 20\}$. The regression algorithm used is a random forest with 25 trees grown for each forest.

The results are reported in Tables 6, 7 and 8. The coverage level for the proposed method oscillates between levels $1 - 2\alpha$ and $1 - \alpha$. The smaller sets are obtained on average by split conformal prediction with a miscoverage rate equal to $2\alpha$. The suggested methods improve quite significantly the performance of cross-conformal prediction in terms of set size, although the variability is generally higher. There is a slight decrease in the size of sets, and a slight increase in variability, of the `e` and `eu-mod-cross` methods as $K$ increases. The results for the split conformal conformal prediction are essentially the same for all tables as a different number of folds is applicable just for cross conformal prediction and its variants.

*Table 6.* Results for the "Abalone dataset" using random forest as regression algorithm. The results refer to set size, except for the last row, which refers to the marginal coverage. The number of folds is $K = 5$ and the $\alpha$-level is set to 0.1. The marginal coverage of the proposed methods lies between $1 - 2\alpha$ and $1 - \alpha$.

|          | cross | e-cross | u-cross | eu-cross | split | split $(2\alpha)$ |
|----------|-------|---------|---------|----------|-------|-------------------|
| Mean     | 6.864 | 6.372   | 5.708   | 5.477    | 6.730 | 4.617             |
| Sd       | 0.074 | 0.255   | 0.040   | 0.316    | 0.155 | 0.103             |
| Median   | 6.858 | 6.332   | 5.713   | 5.390    | 6.703 | 4.583             |
| Min      | 6.798 | 6.049   | 5.659   | 5.195    | 6.512 | 4.516             |
| Max      | 6.986 | 6.762   | 5.755   | 6.015    | 6.912 | 4.777             |
| Coverage | 0.911 | 0.890   | 0.873   | 0.854    | 0.903 | 0.803             |

*Table 7.* Results for the "Abalone dataset" using random forest as regression algorithm. The results refer to set size, except for the last row, which refers to the marginal coverage. The number of folds is $K = 10$ and the $\alpha$-level is set to $0.1$. The marginal coverage of the proposed methods lies between $1 - 2\alpha$ and $1 - \alpha$.

|          | cross | e-cross | u-cross | eu-cross | split | split $(2\alpha)$ |
|----------|-------|---------|---------|----------|-------|-------------------|
| Mean     | 6.869 | 6.323   | 5.695   | 5.444    | 6.816 | 4.741             |
| Sd       | 0.060 | 0.225   | 0.069   | 0.258    | 0.171 | 0.110             |
| Median   | 6.865 | 6.360   | 5.692   | 5.450    | 6.815 | 4.750             |
| Min      | 6.725 | 5.559   | 5.545   | 4.667    | 6.351 | 4.419             |
| Max      | 7.030 | 6.800   | 5.878   | 6.061    | 7.235 | 5.060             |
| Coverage | 0.910 | 0.886   | 0.863   | 0.843    | 0.899 | 0.798             |

*Table 8.* Results for the "Abalone dataset" using random forest as regression algorithm. The results refer to set size, except for the last row, which refers to the marginal coverage. The number of folds is $K = 20$ and the $\alpha$-level is set to $0.1$. The marginal coverage of the proposed methods lies between $1 - 2\alpha$ and $1 - \alpha$.

|          | cross | e-cross | u-cross | eu-cross | split | split $(2\alpha)$ |
|----------|-------|---------|---------|----------|-------|-------------------|
| Mean     | 6.855 | 6.210   | 5.649   | 5.379    | 6.817 | 4.717             |
| Sd       | 0.062 | 0.340   | 0.063   | 0.381    | 0.190 | 0.089             |
| Median   | 6.850 | 6.283   | 5.642   | 5.403    | 6.837 | 4.728             |
| Min      | 6.648 | 4.992   | 5.520   | 4.321    | 6.396 | 4.503             |
| Max      | 7.032 | 6.760   | 5.787   | 6.315    | 7.358 | 4.971             |
| Coverage | 0.909 | 0.879   | 0.862   | 0.839    | 0.899 | 0.795             |

**Electricity consumption dataset.** We additionally analyze a real dataset on electricity consumption, where accurate uncertainty quantification is crucial as the supplier's revenue depends on customer energy use. The dataset contains $35\,411$ observations and 20 covariates; $30\,000$ observations are used for training, while the remaining ones are used for testing. The splitting is repeated 100 times, and the algorithm used is a random forest (with `ntree = 100`). In this case, methods such as full conformal prediction or jackknife+ can be computationally expensive due to the relatively large number of observations, making sample-splitting-based methods preferable.

The results are presented in Table 9. The column "uneven split" represents split conformal prediction, where the training set comprises a $(K - 1)/K$ fraction of the data points. This corresponds to split conformal prediction with the same fraction of training data points used by a single round of cross conformal prediction. The last two columns represent cases where the p-values are combined using the Bonferroni rule at level $\alpha$ (i.e., $K \min(\mathbf{p})$) or at level $2\alpha$ (i.e., $K/2 \min(\mathbf{p})$) rather than the simple average. This implies that the corresponding prediction sets have coverage at least equal to $1 - \alpha$ and $1 - 2\alpha$, respectively.

The smaller sets are obtained by the `eu-mod-cross` method, and in general there is an improvement in the set size if we use cross conformal prediction instead of split conformal prediction. Bonferroni method produces very large prediction sets and this is expected since the Bonferroni rule is not powerful when p-values (or sets) are highly dependent. Indeed, the Bonferroni correction is tightest when the p-values are nearly independent, while the conformal p-values across folds are highly dependent. On the other hand, the rule "twice the mean" used in cross conformal prediction is more powerful when p-values are dependent; see Section 6.1 Vovk & Wang (2020) for a discussion. This aligns with Theorem 4 in Lei et al. (2018), which states that the use of multisplit conformal prediction in conjunction with the Bonferroni rule produces wide sets (specifically, under some assumptions, sets are wider than "single split" conformal prediction). The average set size of "uneven" split conformal prediction is smaller than that of split conformal prediction but slightly larger than that of cross conformal prediction. Moreover, its variability (Sd) is higher.

*Table 9.* Results for the "Electricity consumption dataset" using random forest as regression algorithm. The results refer to set size, except for the last row, which refers to the marginal coverage. The number of folds is $K = 10$ and the $\alpha$-level is set to 0.1. The marginal coverage of the proposed methods lies between $1 - 2\alpha$ and $1 - \alpha$. The Bonferroni method gives large prediction sets.

|  | mod-cross | e-mod-cross | u-mod-cross | eu-mod-cross | cross | split | uneven split | split $(2\alpha)$ | Bonf | Bonf $(2\alpha)$ |
|---|---|---|---|---|---|---|---|---|---|---|
| Mean | 50.85 | 47.10 | 33.78 | 32.27 | 50.71 | 59.59 | 52.25 | 26.52 | 196.68 | 149.63 |
| Sd | 0.49 | 1.66 | 0.31 | 1.54 | 0.49 | 1.64 | 2.97 | 0.54 | 6.51 | 2.92 |
| Median | 50.83 | 47.38 | 33.80 | 32.47 | 50.69 | 59.64 | 52.12 | 26.47 | 197.46 | 149.71 |
| Min | 49.73 | 41.97 | 32.99 | 27.74 | 49.60 | 56.54 | 44.45 | 25.25 | 174.87 | 141.06 |
| Max | 52.38 | 49.67 | 34.41 | 36.20 | 52.25 | 64.95 | 59.84 | 27.96 | 210.94 | 157.85 |
| Coverage | 0.91 | 0.89 | 0.86 | 0.85 | 0.90 | 0.90 | 0.90 | 0.80 | 0.98 | 0.97 |

# F. Proofs of the Results

*Proof of Proposition 4.1.* Let $G : (\mathcal{X} \times \mathcal{Y})^{n+1} = \mathcal{Z}^{n+1} \to [0, 1]^K$ be the transformation that takes as input the $n + 1$ iid (and thus exchangeable) data points $Z_1, \ldots, Z_{n+1}$ and returns as output the p-values $P_1(Y_{n+1}), \ldots, P_K(Y_{n+1})$. In other words, the $i$-th element of $G$ is computed by training the algorithm $\mathcal{A}$ using the dataset $\mathcal{D}_{[n] \setminus I_i}$ and then computing the scores and the corresponding p-value defined in (5) using data points in $\mathcal{D}_{I_i \cup \{n+1\}}$. It is important to note that the score function $s$ satisfies the condition in (1), and so the scores do not depend on the order of the data points in $\mathcal{D}_{[n] \setminus I_i}$. Let $\sigma_1 : [n] \to [K]$ be the function that assigns the training data points to the $K$ different folds and $\sigma_2 : [n] \to [m]$ be the function that assigns the positions of the training data points within the assigned folds. In words, each point $i \in [n]$ is assigned a unique pair $\{\sigma_1(i), \sigma_2(i)\}$ that identifies its fold and its position inside the fold. For example, if $\sigma_1(1) = 2$ and $\sigma_2(1) = 3$ then the first data point in the original dataset is the third data point in the second fold. Let $\pi_1 : [K] \to [K]$ be a permutation of the indices, then for all $z \in \mathcal{Z}^{n+1}$,

$$\pi_1 G(z_1, \ldots, z_n, z_{n+1}) = G(\pi_2(z_1, \ldots, z_n, z_{n+1})),$$

where $\pi_2 : [n + 1] \to [n + 1]$ is such that

$$\pi_2(i) = \begin{cases} [\pi_1(\sigma_1(i)) - 1] \cdot m + \sigma_2(i), & i \neq n + 1, \\ n + 1, & i = n + 1. \end{cases}$$

In words, $\pi_2$ permutes the training data points into their respective permuted folds (i.e., $i \in I_{\pi_1(\sigma_1(i))}$), while the test point remains in the $(n + 1)$-th position. It holds that $G(\cdot)$ preserves exchangeability and this concludes the proof. □

*Proof of Theorem 4.4.* By definition

$$\min_{\ell \in [K]} \frac{1}{\ell} \sum_{k=1}^{\ell} P_k(y) \leq \frac{1}{K} \sum_{k=1}^{K} P_k(y),$$

so less points will be included in the set. From Proposition 4.1 we have that the conformal p-values are exchangeable. The coverage property in (11) is a direct consequence of the result stated in Gasparin et al. (2025), which states that $\min_{\ell \in [K]}(1/\ell) \sum_{k=1}^{\ell} P_k(Y_{n+1})$ is a valid p-value up to a factor of 2 if p-values $P_1(Y_{n+1}), \ldots, P_K(Y_{n+1})$ are exchangeable. □

*Proof of Theorem 4.6.* By definition

$$\frac{1}{2 - U} \frac{1}{K} \sum_{k=1}^{K} P_k(y) \leq \frac{1}{K} \sum_{k=1}^{K} P_k(y),$$

since $1/(2 - U) \leq 1$ almost surely. The result implies that less points will be included in the set. The coverage property in (13) is a consequence of Gasparin et al. (2025), which states that

$$\frac{2}{2 - U} \frac{1}{K} \sum_{k=1}^{K} P_k(Y_{n+1})$$

is a valid p-value. In particular, the result holds under arbitrary dependence of the starting p-values $P_1(Y_{n+1}), \ldots, P_K(Y_{n+1})$.

$\square$

*Proof of Theorem 4.7.* By definition

$$\min \left\{ \frac{1}{2-U} P_1(y), \min_{\ell \in [K]} \frac{1}{\ell} \sum_{k=1}^{\ell} P_k(y) \right\} \leq \min_{\ell \in [K]} \frac{1}{\ell} \sum_{k=1}^{\ell} P_k(y).$$

The result implies that less points will be included in the set and so $\hat{C}_{n,K,\alpha}^{\text{eu-mod-cross}}(X_{n+1}) \subseteq \hat{C}_{n,K,\alpha}^{\text{e-mod-cross}}(X_{n+1})$. The fact that $\hat{C}_{n,K,\alpha}^{\text{e-mod-cross}}(X_{n+1}) \subseteq \hat{C}_{n,K,\alpha}^{\text{mod-cross}}(X_{n+1})$ is outlined in Theorem 4.4.

From Proposition 4.1 we have that the conformal p-values are exchangeable. The coverage property in (15) is a consequence of the fact that

$$\min \left\{ \frac{1}{2-U} P_1(Y_{n+1}), \min_{\ell \in [K]} \frac{1}{\ell} \sum_{k=1}^{\ell} P_k(Y_{n+1}) \right\}$$

is a p-value up to a factor of two if p-values $P_1(Y_{n+1}), \ldots, P_K(Y_{n+1})$ are exchangeable (Gasparin et al., 2025, Appendix B).

$\square$

*Proof of Theorem 4.9.* Comparing Equation (6) with the set defined in Equation (8) we have that $\hat{C}_{n,K,\alpha'}^{\text{mod-cross}}(X_{n+1})$ coincides with $\hat{C}_{n,K,\alpha}^{\text{cross}}(X_{n+1})$. The same result is obtained, for example, in Vovk et al. (2022a, Chapter 4.4). The coverage statement and the properties regarding the size of the sets are corollaries of Theorems 4.4, 4.6 and 4.7, applied with threshold $\alpha'$.

$\square$

