# OpenReview forum: "Improving the Statistical Efficiency of Cross-Conformal Prediction"
_ICML.cc/2025/Conference — ICML 2025 poster_

### Official Review · Reviewer_BC25 · 2025-03-10

**Overall Recommendation:** 4

**Summary:**

This paper proposes several new variants of (modified) cross-conformal prediction--called  e/u/eu-modified cross conformal prediction--that theoretically and empirically attain more efficient (i.e., smaller and thus more informative) prediction sets/intervals than the original (modified) cross-conformal prediction method, all while maintaining the same worst-case coverage guarantee ($\geq 1-2\alpha$, where $\alpha \in (0, 1)$ is the target miscoverage rate). These new methods and their guarantees are derived using new results on the combination of exchangeable p-values in Gasparin et al. 2024.

## update after rebuttal

I maintain my positive score--I appreciate the authors' clarifications, highlighting experimental results related to coverage variability, and discussion related to my question about recommendations on conditions for using which CP methods (which could be good to mention in the paper's discussion/conclusion). Although much of the theoretical analysis is mainly citing Gasparin et al. 2024, the contribution is solid, and with the paper's clear writing, it should be of interest to the community.

**Claims And Evidence:**

Yes. Proofs are provided for the main theorems and sufficiently convincing experiments are provided on both synthetic data with an unstable algorithm and real data.

**Essential References Not Discussed:**

Essential references are discussed. A few further related works that could be relevant or interesting for the authors to look at or mention are given in the “Other Comments or Suggestions” section.

**Experimental Designs Or Analyses:**

Mostly yes, I have reviewed the overall experimental setting described and they seem reasonable and reliable. Eg, the synthetic-data experiments are reasonably implemented with the same setting as Barber et al. 2021, where the algorithm is unstable around d=80 which makes for an important “adversarial” test case where it is good to verify coverage claims (and coverage appears appropriate). The real data experiments include standard tabular UCI datasets that are often used for evaluation in the conformal literature.

Some suggestions:
- It would be good to make it clearer in the figure captions or figures themselves what alpha is (ie, what target coverage is)
- (Optional) It could be interesting/valuable to add an evaluation of coverage *variance*, ie how the coverage varies over different random draws of training/cal data.
- (Optional) It could be interesting/valuable to add supplemental experiments comparing the proposed e/u/eu-mod-cross method run with target coverage *$\alpha/2$* versus cross/split conformal run at target coverage $\alpha$. That is, especially since the empirical coverage of the proposed methods may dip below the target level (ie in $[1-2\alpha, 1-\alpha]$), practitioners may wish to run eg the eu-mod-cros method targeting $1-\alpha/2$ to ensure $\geq1-\alpha$ worst-case miscoverage. So, further discussion and evaluation of this would be useful.

**Methods And Evaluation Criteria:**

Yes. Sufficiently convincing experiments are provided on synthetic data with an unstable algorithm (ie, instability around d=80 is an “adversarial” case where is important to show coverage holds empirically) and on real datasets, supporting the provided coverage guarantees and claims about more efficient prediction sets.

**Other Comments Or Suggestions:**

**Other refs that may be relevant/of interest to authors:**

*Other ref using result of Vovk and Wang (2020) for combining conformal sets:*
- Stutz, D., Roy, A. G., Matejovicova, T., Strachan, P., Cemgil, A. T., and Doucet, A. Conformal prediction under ambiguous ground truth. arXiv preprint arXiv:2307.09302, 2023
- Couple refs on selecting conformal sets for efficiency:
- Yang, Y. and Kuchibhotla, A. K. Selection and aggregation of conformal prediction sets. Journal of the American Statistical Association, pp. 1–13, 2021.
- Liang, R., Zhu, W., & Barber, R. F. (2024). Conformal prediction after efficiency-oriented model selection. arXiv preprint arXiv:2408.07066.

*Refs on related cross-validation style conformal methods (jackknife+ and CV+) under distribution shift (ie, extending the proposed methods to account for distribution shift could be an interesting future direction for the authors):*
-  Prinster, D., Liu, A., & Saria, S. (2022). Jaws: Auditing predictive uncertainty under covariate shift. Advances in Neural Information Processing Systems, 35, 35907-35920.
- Prinster, D., Saria, S., & Liu, A. (2023, July). Jaws-x: addressing efficiency bottlenecks of conformal prediction under standard and feedback covariate shift. In International Conference on Machine Learning (pp. 28167-28190). PMLR.


**Other Suggestions:**
- Related to “Weakness” mentioned before, it could be useful to add discussion about when the authors recommend practitioners to use one method or another. Eg, when/why would be recommended to use proposed methods at target $1-\alpha$ (thus achieving $1-2\alpha$ guarantee) versus at target $1-\alpha/2$ (thus achieving $1-\alpha$ guarantee).
- It may be helpful to use language such as “target/nominal” coverage or miscoverage when referring to the user’s inputted $1-\alpha$ or $\alpha$, to more clearly distinguish from worst-case guarantee level.
- Authors may want to consider slight modification to names of their methods: Eg, it’s understandable from reading the paper why they name one of their methods “exchangeable modified cross-conformal prediction (e-mod-cross),” ie because it’s attained using results on merging exchangeable p-values, but it may cause confusion, as sometimes standard CP methods are called “exchangeable CP” methods to refer to assumption of exchangeable data. Example slight modification could be “exchangeable p-value modified cross conformal (ep-mod-cross)”
- It might be advisable to state exchangeability/IID as an assumption within the actual theorem statements themselves
- End of proof of Theorem 4.6: When you state “holds under arbitrary dependence,” do you mean arbitrary dependence that still maintains exchangeability? If so, would be good to make this explicit to ensure avoiding confusion (understand may be implicit)

**Other Strengths And Weaknesses:**

*Strengths:* The paper is very clearly written, and all claims are sufficiently supported with proofs and/or empirical evidence. The paper’s contribution is framed appropriately and can be useful for future progress on thinking about improving the efficiency of conformal prediction sets.

*Weaknesses:* One limitation of the proposed methods is that--whereas the original (modified) cross-conformal methods typically have empirical coverage at or above the target level ($\geq 1-\alpha$)--the proposed methods seem to have typical empirical coverage that may be below the target level and closer to the worst-case guarantee, ie, $\in [1-2\alpha, 1-\alpha]$ (eg, see Table 2). It would probably improve the paper to further discuss and/or evaluate this, and potentially to provide recommendations for how the proposed method should be used in practice: ie, should practitioners run the method targeting a more conservative level (eg, targeting $1-2\alpha$) to achieve $1-\alpha$ in the worst-case, or run targeting $1-\alpha$, while acknowledging that empirically it appears more likely that the coverage will fall in $[1-2\alpha, 1-\alpha]$?

**Questions For Authors:**

No major questions, see “Suggestions” for minor questions/comments. Congrats on a nice paper!

**Relation To Broader Scientific Literature:**

*Most relevant prior literature:* Cross-conformal prediction was originally introduced by Vovk (2015) with worst-case coverage guarantees of the form $1-2\alpha - B$ (for some $B=2(1-\alpha)(K-1)/(n+K)$ that becomes negligible for smaller numbers of folds, $K$); a simple modification was introduced in Vovk et al. (2018), which cites earlier work by Vovk and Wang (2012), to achieve a guarantee at $\geq 1-2\alpha$. Barber et al. (2021) also considers this “modified cross-conformal” method from Vovk et al. (2018) (to introduce a related cross-validation+ method). Gasparin et al. (2024) has results on combining exchangeable p-values that are used in this paper.

*Contribution in context:* The proposed method improves the statistical efficiency of prior cross-conformal methods in that it empirically attains sharper (smaller and more informative) prediction sets, and theoretically the sets are no larger than original cross-conformal. The main proof steps are largely citing Gasparin et al. (2024).

**Theoretical Claims:**

Yes. I have made an effort to check the proofs and they appear to be sound--key steps largely cite results in Gasparin et al. 2024. The theorems provide coverage guarantees for the proposed methods (Theorems 4.4, 4.6, & 4.7) and demonstrate that the proposed methods are not more conservative (ie, prediction sets no larger) than the original cross-conformal method (Theorem 4.9).

---

> ### Author Rebuttal · Authors · 2025-03-31
>
> Thank you very much for your positive and constructive comments on our paper. Please find below a detailed response to your questions.
>
> ### Experimental Designs:
> - To improve clarity, we will add the target coverage level $1-\alpha=0.9$ in the captions.
> - As index of variability, in Table 1, we report the maximum and the minimum of the empirical coverage observed over the 20 replications. A part of the table (which refers to LM) is reported below. The variability (see Range) is not so different for all the methods.
>
> || mod-cross | e-mod-cross | u-mod-cross | eu-mod-cross | cross | split | split(2$\alpha$) |
> |--|--|--|--|--|--|--|--|
> | Mean | 0.903 | 0.899 | 0.851 | 0.858 | 0.902 | 0.902 | 0.800 |
> | Min | 0.896 | 0.885 | 0.834 | 0.840 | 0.895 | 0.890 | 0.779 |
> | Max | 0.917 | 0.915 | 0.874 | 0.881 | 0.916 | 0.920 | 0.831 |
> | Range | 0.021 | 0.030 | 0.040 | 0.041 | 0.021 | 0.030| 0.052 ||
>
> - Some of the simulation studies report also the results of split conformal prediction with target coverage $1-2\alpha$. This implies that the proposed variants (trained at level $\alpha$) and split conformal prediction with target level $1-2\alpha$ guarantee the same teorethical coverage (and this is equivalent of using $\alpha/2$ to obtain coverage $1-\alpha$). In Table 5, for example, the eu-mod-cross method is better in terms of size than split conformal prediction trained at level $2\alpha$. (Standard) cross-conformal prediction at level $2\alpha$ could be added; however, we expect smaller sets if compared to those of split conformal but with the same empirical coverage (as observed for the $\alpha$ level). We discuss more on this in the first point of *Other suggestion*.
>
> ### Weaknesses:
> See the first point in *Other Suggestions*.
>
> ### References:
> Thank you very much for pointing out some relevant references. We will add them to the paper.
>
> ### Other Suggestions:
> - Thank you for you comment. The question of which variant of cross-conformal prediction is to be preferred in practice is subtle:
>     - If one really needs to have a rigorous $1-\alpha$ guarantee against the worst case distributions and unstable algorithms (for example, if there are downstream decisions that crucially depend on the provided theoretical guarantee), then it makes sense to run our new methods at level $\alpha/2$.
>     - If one is only using conformal prediction as "weak guidance" for downstream decisions, and the user is somehow ok with violations of the target $1-\alpha$, then we recommend running our variants of cross-conformal at level $\alpha$ (where its worst case guarantee is $1-2\alpha$ but it achieves coverage in between $1-2\alpha$ and $1-\alpha$).
>     - If the situation is in between, where conformal prediction is neither being used extremely rigorously, nor as loose guidance, but one wants $1-\alpha$ coverage for "typical" datasets and algorithms, but is ok with both overcoverage and undercoverage for odd distributions/algorithms, then perhaps the original (modified) cross-conformal is best.
>     - If one does not tolerate over- or under-coverage of any sort, and really wants essentially exact $1-\alpha$ coverage, then only split conformal delivers the goods.
>
> - We agree with this suggestion and we will revise the text accordingly to clarify the distinction.
>
> - Thank you for the suggestion, we will consider this change.
>
> - We will add the assumption of exchangeability within the theorem statements, as suggested. Example:
> "*... . If data are exchangeable,*
> $$
> P\left(Y_{n+1}\in\hat C_{n,K,\alpha}^{\mathrm{e-mod-cross}}(X_{n+1})\right)\ge1-2\alpha.
> $$
>
> - Theorems 4.4 and 4.7 require that the p-values $P_1, \dots, P_K$ be exchangeable (and the exchangeability of the p-values is shown in Theorem 4.2). However, the assumption of exchangeability is not necessary for proving Theorem 4.6. It is sufficient that $P_1, \dots, P_K$ are valid p-values (i.e., $P(P_k \leq \alpha) \leq \alpha$), and this assumption is satisfied in the case of rank-based p-values. In fact, the rules $2 \times \text{mean}(\mathbf{P})$ (used by Vovk et al. (2018) to prove the coverage guarantee for cross conformal prediction) and $2/(2-U) \times \text{mean}(\mathbf{P})$ are valid under arbitrary dependence, which is a broader condition than exchangeability. We will clarify this in the text.

---

### Official Review · Reviewer_opLB · 2025-03-18

**Overall Recommendation:** 4

**Summary:**

The authors move from the interesting, yet relatively limited literature on methods that work on improving the dramatic data inefficiency of split conformal prediction.
Their proposal exploits novel results about p-value combination to introduce a variant of the well known cross-conformal prediction methods that achieves better theoretical coverage properties.
After a thorough description, and proofs about its theoretical properties, the author put their modified cross-conformal prediction method to the test, in both a simulation and a real world example.

**Claims And Evidence:**

All claims are well supported by well crafted proofs, I am a bit more skeptical about the applicative results (but more on this later)

**Essential References Not Discussed:**

I would mention the second edition of Algorithmic Learning in a Random World (Vovk et al. 2023).
Lei et al. 2018 moreover specifies a very interesting results about merging multiple splits in Conformal Prediction, which triggers some questions (see below)

**Experimental Designs Or Analyses:**

As mentioned previously, I believe that the experimental study on real world cases is a bit too limited, as it does not really give much insights with respect to the situations where the method proposed by the authors give significant practical advantages

**Methods And Evaluation Criteria:**

Both the simulation and the real world example are satisfactory. It would be worth maybe to propose a slightly extended real world test, in order to grasp better the practical advancements introduced by the method.

**Other Comments Or Suggestions:**

nothing of relevance.

**Other Strengths And Weaknesses:**

nothing of relevance

**Questions For Authors:**

I find the conclusions quite underwhelming. I appreciate the interesting work the authors have proposed, but fail to understand the practical implications of it. Is the method useful? In what situations? In what use cases? Please give more insights.

I find this result contrasting quite strikingly with a result offered in Lei et al. 2018 (section 2.3). I wonder if you have something to comment on this.

**Relation To Broader Scientific Literature:**

The background on Conformal Prediction is not correctly stated. CP was not introduced in Vovk et al. 2005, (which instead represents an early formalisation of the first research on the subject), but in Saunders et al 1999 (https://www.ijcai.org/Proceedings/99-2/Papers/010.pdf).

**Theoretical Claims:**

I have carefully checked all the proofs in the paper, and everything seems in order.

---

> ### Author Rebuttal · Authors · 2025-03-31
>
> We appreciate your feedback. Below is our response to your comments
> # Experimental Designs
> We now additionally analyze a real dataset on electricity consumption, where accurate uncertainty quantification is crucial as the supplier’s revenue depends on customer energy use. The dataset contains 35,411 observations and 20 covariates; 30,000 observations are used for training, while the remaining ones are used for testing. The splitting is repeated 20 times, and the algorithm used is a random forest. In this case, methods such as Full CP and Jackknife+ can be computationally expensive, making sample-splitting-based methods preferable. From the table, it can be seen that Cross CP is more efficient than Split CP, producing more informative sets while ensuring valid coverage. See the first point in *Questions* below for a further discussion.
>
> The column *Uneven split* represents Split CP, where the training set comprises a $(K-1)/K$ fraction of the data points (ie, Split CP with the same fraction of training data points used by a single round of Cross CP). The last two columns represent cases where p-values are merged using the Bonferroni rule at level $\alpha$ (ie,$K\min(p)$) or at level $2\alpha$ (ie,$K/2\min(p)$). These results are discussed in the 2nd point of the *Questions* section
> ||mod-cross|e-mod-cross|u-mod-cross|eu-mod-cross|cross|split|uneven split|split(2$\alpha$)|Bonf|Bonf(2$\alpha$)|
> |-|-|-|-|-|-|-|-|-|-|-|
> |Mean|50.53|47.20|33.64|32.26|50.40|59.59|53.43|26.59|223.86|153.81|
> |Sd|0.52|1.70|0.32|1.27|0.51|1.39|2.74|0.45|10.99|5.82|
> |Min|49.50|42.74|33.00|29.08|49.36|57.09|48.12|25.64|202.48|141.66|
> |Max|51.47|49.23|34.43|33.72|51.34|62.26|58.62|27.71|244.33|165.03|
> |Cov|0.90|0.89|0.86|0.84|0.90|0.90|0.90|0.80|0.98|0.96|
> # References
> - Thank you for pointing to this. We will add the paper by Saunders et al
> - We will discuss the two references appropriately in the text
> # Questions
> **Conclusions**
>
> We believe that the methods has practical utility. For instance, when dealing with a large number of observations (and potentially a high-dimensional feature space), some methods, such as full CP and jackknife+, become impractical. One possible approach is to trade off some statistical efficiency for reduced computational cost by using methods based on sample-splitting. However, split CP can be inefficient because it only uses a subset of the data for model training. Cross CP generally improves the efficiency of split CP but can overcover. Our work improves statistical efficiency of Cross CP (at the same computational efficiency) while maintaining its coverage guarantee.
>
> The question of which variant of cross CP is to be preferred in practice is subtle.
> - If one really needs to have a rigorous $1-\alpha$ guarantee against the worst case distributions and unstable algorithms (for example, if there are downstream decisions that crucially depend on the provided theoretical guarantee), then it makes sense to run our new methods at level $\alpha/2$
> - If one is only using conformal prediction as "weak guidance" for downstream decisions, and the user is somehow ok with violations of the target $1-\alpha$, then we recommend running our variants of cross-conformal at level $\alpha$ (where its worst case guarantee is $1-2\alpha$ but it achieves coverage in between $1-2\alpha$ and $1-\alpha$)
> - If the situation is in between, where conformal prediction is neither being used extremely rigorously, nor as loose guidance, but one wants $1-\alpha$ coverage for "typical" datasets and algorithms, but is ok with both overcoverage and undercoverage for odd distributions/algorithms, then perhaps the original (modified) cross-conformal is best
> - If one does not tolerate over- or under-coverage of any sort, and really wants essentially exact $1-\alpha$ coverage, then only split conformal delivers the goods
>
> We acknowledge that the conclusions can be improved and will incorporate the above takeaway messages there
>
> **Discussion Lei et al**
>
> The results in Lei et al are based on the Bonferroni rule, that is not powerful when p-values (or sets) are highly dependent. Indeed, the Bonferroni correction is tightest when the p-values are nearly independent, while the conformal p-values across folds are highly dependent. On the other hand, the rule "twice the mean" used in cross CP is more powerful if p-values are dependent; see Sec.6.1 in Vovk & Wang (2020). For completeness; the above table also reported the use of the Bonferroni rule as merging function. The last 2 columns guarantee coverage of at least $1-\alpha$ and $1-2\alpha$, respectively. However, the methods have coverage near 1. This aligns with Thm.4 in Lei et al, which states that multisplit+Bonferroni produce wide sets (specifically, sets are wider than single split CP). For comparison, we have added split CP with an uneven split. The set size is smaller than that of Split CP but larger than that of Cross CP and variants. Moreover, its Sd is higher. We will add this discussion

---

> > ### Comment · Reviewer_opLB · 2025-04-02
> >
> > I thank the authors for the insightful answers, which I believe are very satisfactory.
> > I now sincerely believe the paper to be of better quality, and to be more useful and clear in terms of clarifying the issues raised in the official comment
> > I will increase my evaluation

---

### Official Review · Reviewer_uMcf · 2025-03-19

**Overall Recommendation:** 4

**Summary:**

The paper proposes new variants of cross-conformal prediction to obtain smaller prediction sets while guaranteeing the same worst-case miscoverage rate. They use recent results on the combination of dependent and exchangeable p-values to obtain their results. Empirical evaluation on simulated data and the news popularity dataset show that their proposed improvements lead to smaller sets than cross-conformal prediction and modified cross-conformal prediction. Similar improvements are observed in the additional experiments.

**Claims And Evidence:**

Claims made in the paper are supported by theory (Theorem 4.4 - 4.9) and empirical evidence. The results show that the proposed improvements result in smaller prediction sets while maintaining the coverage guarantees.

**Essential References Not Discussed:**

I feel the paper is fairly complete in its discussion of important references for understanding the context.

**Experimental Designs Or Analyses:**

I checked the soundness of all experiments, including the simulation study, real data application, and additional results reported in the appendix. The experimental details and choices have been carefully explained, and there is effort to experiment with different algorithms and parameters.

**Methods And Evaluation Criteria:**

The empirical evaluation is extensive and the experiments are performed on multiple datasets. The benchmark datasets and the reported size and coverage metrics make sense for the problem.

**Other Comments Or Suggestions:**

minor typo: p2 l68: obtained ‘by’ applying

**Other Strengths And Weaknesses:**

I appreciate the clear writing and how the paper lays out past work to contextualize their work and contributions better. The remarks in the paper further add to the clarity.

**Questions For Authors:**

No specific questions.

**Relation To Broader Scientific Literature:**

Building on the literature of cross-conformal prediction and combination of dependent p-values, this paper contributes in producing smaller sets than cross-conformal prediction (and modified version) and split conformal prediction; while the empirical coverage lies between $1 - 2\alpha$ and $1 - \alpha$. The guarantees presented and supporting empirical evidence show improvement over prior work with similar model training cost.

**Theoretical Claims:**

I went over the proofs for Theorem 4.6-4.9 in Section F.

---

> ### Author Rebuttal · Authors · 2025-03-31
>
> Thank you very much for your positive feedback on our paper. We truly appreciate your comments.
>
> Thank you for noticing the typo, we corrected it.

---

### Official Review · Reviewer_qkMw · 2025-03-26

**Overall Recommendation:** 3

**Summary:**

The authors propose new variants of cross-conformal prediction that leverages recent results on combining p-values through exchangeability and randomization. They theoretically demonstrate that their methods can reduce the size of the prediction set while maintaining a marginal coverage of at least $1 - 2\alpha$. The paper also highlights the computational advantages of these new methods, as they require training the model only a limited number of times (K times) rather than for every possible response value.

**Claims And Evidence:**

The theoretical guarantees are well supported by proofs in the appendices, leveraging the coverage properties of p-value combination results. Simulations and real-world experiments validate the smaller prediction set sizes compared to baselines, with empirical coverage aligning with theoretical bounds.

**Essential References Not Discussed:**

NA

**Experimental Designs Or Analyses:**

The experimental design is appropriate but could be strengthened by increasing the number of trials (e.g., from 20 to 100) to improve reliability and including the experiments with varying fold sizes.

**Methods And Evaluation Criteria:**

The use of exchangeable p-values and randomization is novel and logically derived in this context. The regression tasks on benchmark datasets are appropriate.

**Other Comments Or Suggestions:**

No further suggestions.

**Other Strengths And Weaknesses:**

Strengths:
1.	The research innovatively utilizes combination of p-values through exchangeability and randomization to cross-conformal prediction.
2.	Maintains theoretical guarantees with smaller prediction sets and computational efficiency (K folds).
Weaknesses:
1.	Randomization introduces variability in prediction sets, which may limit deterministic applications and lead to reproducibility concerns.

**Questions For Authors:**

NA

**Relation To Broader Scientific Literature:**

The work in this paper builds on cross-conformal prediction and recent p-value combination methods, improving the statistical efficiency, and further advancing the development of conformal prediction.

**Theoretical Claims:**

The theoretical claims are well supported by the proof in appendices.

---

> ### Author Rebuttal · Authors · 2025-03-31
>
> Thank you for your feedback on our paper. Please find below a detailed response to your concerns.
>
> ### Experimental Designs Or Analyses:
> Thank you for the suggestions. Below, we present an example where the experimental setting is identical to the last experiment in Appendix E, but with $K$ set to 5, 10, and 20. In addition, the number of trials is 100. The second table ($K=10$) is similar to the table reported in the paper with 20 trials (this number is used in many conformal prediction works; see, for example, Barber et al. (2021) or Romano et al. (2019)). There is a slight decrease in the empirical size of sets, and a slight increase in variability, of the e and eu-mod-cross methods as $K$ increases. Clearly, nothing changes for the split conformal methods. We will add the example and add more trials for other experiments.
>
> | K=5 | cross | e-cross | u-cross | eu-cross | split | split($2\alpha$) |
> |--|--|--|--|--|--|--|
> | Mean | 6.887 | 6.382 | 5.711 | 5.508 | 6.764 | 4.735 |
> | Sd | 0.072 | 0.162 | 0.058 | 0.182 | 0.167 | 0.100 |
> | Median | 6.881 | 6.389 | 5.711 | 5.518 | 6.748 | 4.719 |
> | Min | 6.656 | 5.964 | 5.576 | 5.074 | 6.331 | 4.516 |
> | Max | 7.064 | 6.762 | 5.837 | 6.015 | 7.268 | 4.967 |
> | Coverage | 0.910 | 0.885 | 0.867 | 0.845 | 0.897 | 0.800 |
>
> | K=10 | cross | e-cross | u-cross | eu-cross | split | split(2$\alpha$) |
> |--|--|--|--|--|--|--|
> | Mean | 6.869 | 6.323 | 5.695 | 5.444 | 6.816 | 4.741 |
> | Sd | 0.060 | 0.225 | 0.069 | 0.258 | 0.171 | 0.110 |
> | Median | 6.865 | 6.360 | 5.692 | 5.450 | 6.815 | 4.750 |
> | Min | 6.725 | 5.559 | 5.545 | 4.667 | 6.351 | 4.419 |
> | Max | 7.030 | 6.800 | 5.878 | 6.061 | 7.235 | 5.060 |
> | Coverage | 0.910 | 0.886 | 0.863 | 0.843 | 0.899 | 0.798 |
>
> | K=20 | cross | e-cross | u-cross | eu-cross  | split | split($2\alpha$) |
> |--|--|--|--|--|--|--|
> | Mean | 6.855 | 6.210 | 5.649 | 5.379 | 6.817 | 4.717 |
> | Sd | 0.062 | 0.340 | 0.063 | 0.381 | 0.190 | 0.089 |
> | Median | 6.850 | 6.283 | 5.642 | 5.403 | 6.837 | 4.728 |
> | Min | 6.648 | 4.992 | 5.520 | 4.321 | 6.396 | 4.503 |
> | Max | 7.032 | 6.760 | 5.787 | 6.315 | 7.358 | 4.971 |
> | Coverage | 0.909 | 0.879 | 0.862 | 0.839 | 0.899 | 0.795 |
>
> We additionally analyze the case where the number of observations in each fold differs. Since in cross-conformal prediction the $K$ folds serve the same role (being used for both training and calibration), having folds with significantly different sizes would be unwise. This principle applies not only to cross-conformal prediction but also to other methods that randomly partition the data into $K$ folds, such as Hulc (Kuchibhotla et al., 2024) and MoM (Lugosi & Mendelson, 2019). As explained at the beginning of Section 3, this means that the folds can differ by at most $K-1$ observations (with $K$ typically set to 5 or 10).
> For completeness, we present the results of a simulation study based on the Boston dataset. The setting is the same as described in Appendix E, except that $n=204$ and one fold at random contains 44 observations instead of 40 (a 10% increase). The number of trials is 100. One could, in principle, have four folds with 41 observations and one with 40, but we consider a more 'extreme' setting. The results are presented in the Table, and the conclusions remain qualitatively similar to those reported in Appendix E.
> | | mod-cross | e-mod-cross | u-mod-cross | eu-mod-cross | cross |
> |--|--|--|--|--|--|
> | Mean   | 17.299 | 15.206 | 14.083 | 13.565 | 15.883 |
> | Sd    | 1.542 | 2.218 | 1.211 | 2.159 | 1.414 |
> | Median  | 17.245 | 15.229 | 14.068 | 13.605 | 15.819 |
> | Min   | 14.029 | 9.518 | 11.699 | 8.895 | 13.000 |
> | Max   | 21.196 | 20.869 | 16.885 | 20.199 | 19.243 |
> | Coverage | 0.926 | 0.892 | 0.877 | 0.859 | 0.909 |
>
> ### Weaknesses:
> We agree that randomization can introduce variability in the prediction sets; however, employing randomized and asymmetric combination rules is the only way to enhance the efficiency of cross-conformal prediction sets while preserving the same (worst-case) coverage guarantee. Furthermore, as noted in Remark 4.8, randomization plays a role at various stages of the data pipeline in both cross and split conformal prediction. As discussed in the paper, in some applications of large-scale deployment involving thousands of daily predictions, improved statistical efficiency may be preferred.
>
> In light of the responses provided, we hope we have addressed your concerns and that you may consider raising the score.
>
> Kuchibhotla, A. K., Balakrishnan, S., & Wasserman, L. (2024). The HulC: confidence regions from convex hulls. Journal of the Royal Statistical Society Series B: Statistical Methodology, 86(3), 586-622.
>
> Lugosi, G., & Mendelson, S. (2019). Mean estimation and regression under heavy-tailed distributions: A survey. Foundations of Computational Mathematics, 19(5), 1145-1190.

---

### Decision · Program_Chairs · 2025-05-01

**Decision:**

Accept (poster)

**Comment:**

The paper presents novel methods for cross-conformal prediction that leverage exchangeable p-values and randomization to produce smaller and more efficient prediction sets while maintaining strong theoretical guarantees for worst-case coverage rates. The reviewers agree that the paper makes a significant theoretical and practical contribution, with rigorous proofs and empirical validation on both synthetic and real-world datasets. The methods also demonstrate computational efficiency, requiring fewer model runs compared to prior approaches.

While some reviewers initially suggested improvements, such as providing practical guidance on applying the methods and expanding real-world experiments, these concerns have been thoroughly addressed in the rebuttal. The authors have clarified the choice between targeting 1 − α and 1 − 2α, provided additional context for practical applications, and addressed questions about experimental design. These responses resolve the reviewers' primary concerns and strengthen the paper's overall impact.

Given the strong contributions, clear writing, and effective rebuttal, I recommend accepting the paper. The proposed methods represent a meaningful advancement in the field of conformal prediction and are well-supported by both theory and experiments.